# Not All Who Wander Are Lost: Hallucinations as Neutral Dynamics in Residual Transformers

## Abstract

Hallucinations in autoregressive models arise in two stages: an initial deviation from the truth and its continued propagation during decoding. Existing work addresses the first stage with empirical or diagnostic methods, but there is no fundamental account of the second stage. We give the first structural analysis of how paired continuations of the same prompt evolve inside pre-LayerNorm residual transformers, which form the backbone of most modern LLMs. By examining the residual stack and decoder, we show that their dynamics contain no built-in pull that suppresses deviations and no push that amplifies them. This neutrality is necessary, but not sufficient, for semantic hallucinations: it permits deviations to continue, yet a model can still correct the meaning even when predictive differences persist. Neutrality also yields an explicit upper bound, a separation between deterministic and stochastic effects, and a statistical validation rule at finite sample sizes. A population-level version follows by treating the small deviations across many continuations as agents in a mean-field average, showing that neutrality persists at scale without requiring access to individual weights. Experiments on GPT2 variants and Qwen2.5 models from 0.5B to 3B match the theoretical predictions.

## 1 Introduction

Large language models generate text by predicting the next token, and small errors can accumulate into large departures from the truth. A core challenge is that research on hallucinations is largely empirical. Surveys note the reliance on empirical methods and the resulting uncertainty about full elimination (Huang et al., 2025), and point out that such studies "cannot answer the fundamental question: can hallucination be completely eliminated?" (Xu et al., 2024). Reviews of training pipelines add that "pre-training primarily optimizes for completion," providing no pressure to correct deviations once they arise (Huang et al., 2025). Recent work furthermore notes that LLMs remain largely static after pre-training, with knowledge confined to the immediate context and existing remedies often limited or lacking generalization (Behrouz et al., 2025).

Most direct mitigation strategies focus on onset. Methods such as scheduled sampling (Bengio et al., 2015), sequence-level objectives (Ranzato et al., 2015), retrieval augmentation (Lewis et al., 2020), reinforcement learning with human feedback (Ouyang et al., 2022), and tool use (Schick et al., 2023) reduce the likelihood of the first onset-error by improving grounding or training stability. However, these methods do not specify how a deviation evolves once present, since they leave the internal update rule of the residual backbone untouched.

Diagnostics provide complementary insight. Measures based on inconsistency, contradiction or semantic uncertainty highlight when hallucinations occur (Farquhar et al., 2024; Lin et al., 2021; Manakul et al., 2023; Chen et al., 2024; Mündler et al., 2023), and human evaluation remains important for assessing factuality (Maynez et al., 2020; Kryściński et al., 2020; Ji et al., 2023; Huang et al., 2025). But these methods capture symptoms, not mechanisms, and leave it unclear whether the architecture suppresses deviations, amplifies them, or lets them persist.

This gap has two consequences. First, it obscures why models lose earlier information quickly. If the architecture provides no corrective tendency, then once two continuations differ, the difference can simply persist. This means that nothing in the architecture forces the two paths to reconverge,

allowing deviations introduced at onset to survive step by step. Second, it limits the scope of existing mitigation strategies. Techniques that adjust sampling or provide external tools can influence onset, but if persistence is governed by a structural property of the residual backbone, addressing onset alone cannot prevent deviations from propagating internally.

This leads to the central question of this paper: *Once a deviation is present, what structural law governs how it moves through the residual architecture of a pre-LayerNorm transformer?*

Answering this requires a formal stepwise description of how two slightly different continuations evolve. Section 3 develops the framework and analytical tools for this.

### CONTRIBUTION AND NOVELTY

We show that pre-LayerNorm residual transformers operate under neutral dynamics, a structural property independent of specific learned weights from which we derive three results:

**Architectural neutrality.** Differences between two continuations neither shrink nor grow on average. This identifies the architectural condition under which a deviation can *persist*.

**Predictive control and validation.** Neutrality gives an explicit limit on how far two continuations can move apart at each step and separates the systematic part of this movement from random variation, which enables a test that works with finite samples. Using this test, we provide empirical evidence on GPT2 models that vary in depth and width and on Qwen2.5-0.5B. Our results support the theoretical prediction that neutrality follows from the residual architecture itself and therefore does not depend on scale.

**Population-level behaviour.** A mean-field average over many continuations shows that this same neutral behaviour appears at larger scales without needing access to individual weights.

To our knowledge, this provides the first structural account of persistence in autoregressive transformers. It reframes persistence as an architectural property and makes explicit which parts of hallucination behaviour arise from the backbone dynamics themselves.

## 2 BACKGROUND

Autoregressive transformer decoders generate text one token at a time. At each step the model maintains a hidden state and maps it to a next-token distribution. Small differences in these hidden states can lead to different predictions, and whether such differences persist depends on how the architecture propagates them, which we formalize here.

### 2.1 CONTINUATIONS AND PREDICTIVE BEHAVIOUR

Consider two continuations that start from the same prompt and evolve under the same autoregressive model. At decoding step $t$ the model holds hidden states $(h_t^{(1)}, h_t^{(2)})$, and applies the same decoder to obtain next-token distributions, where $S$ denotes the model's decoder

$$p_t = S(h_t^{(1)}), \qquad q_t = S(h_t^{(2)}).$$

Any difference between $p_t$ and $q_t$ reflects a structural difference between the hidden states. Comparing them provides a direct lens on how the architecture transports small deviations forward across successive decoding steps. To quantify predictive separation we use the Jensen–Shannon divergence $D_t = \mathrm{JS}(p_t, q_t)$ (see Appendix A.2 for properties). In this work $D_t$ is used purely as a structural measure of predictive difference, as it does not measure semantic correctness.

### 2.2 AUTOREGRESSIVE DYNAMICS

Modern transformer decoders, including recent LLMs, commonly use a pre–LayerNorm residual design. In this configuration, LayerNorm is applied before the attention or feed-forward sublayer. This placement is widely adopted because it improves optimization stability and produces well-behaved gradients at depth, as shown in analyses comparing pre– and post–LayerNorm architectures (Xiong et al., 2020; Matarazzo & Torlone, 2025).

A pre–LayerNorm residual block has the form $H_\ell(x) = x + G_\ell(\text{LN}(x))$, where $\text{LN}(x)$ is LayerNorm applied to $x$ including its learnable scale and shift parameters, $G_\ell$ is the sublayer consisting of attention and feedforward components of the transformer, and the residual connection adds the transformed and normalized input back to the original signal.

Composing $L$ blocks yields the residual stack $F = H_L \circ \cdots \circ H_1$. The decoder maps hidden states to predictive distributions through $S(h) = \text{softmax}_T(Wh + b)$, where $Wh + b$ are the pre-softmax scores (logits), $W$ is the learned output projection matrix, and $b$ is the learned bias. The temperature-scaled softmax converts these scores into a distribution over the vocabulary.

Under autoregression a continuation evolves according to

$$h_{t+1} = F(h_t), \qquad p_t = S(h_t), \qquad \tau_t \sim p_t.$$

The combined map $(F, S)$ therefore determines how differences in hidden states are turned into differences in predictive distributions and how these differences propagate across time.

### 2.2.1   SEMANTIC CONVERGENCE VERSUS PREDICTIVE SEPARATION

Because $D_t$ measures the difference between predictive distributions rather than semantic correctness, semantic self–correction does not imply that $D_t$ decreases. Even if the two continuations move toward the same correct answer in meaning, their next–step predictive distributions

$$p_{t+1} = S(h_{t+1}^{(1)}), \qquad q_{t+1} = S(h_{t+1}^{(2)})$$

may still differ. Our analysis therefore concerns how predictive differences propagate through $(F, S)$.

### 2.3   STEPWISE EVOLUTION OF $D_t$

Persistence of any deviation requires that a difference present at step $t$ is not automatically eliminated at step $t+1$. The stepwise evolution of $D_t$ captures exactly this behaviour. At a structural level there are three possible regimes for the map from hidden states to predictive distributions:

- **Contractive:** $D_{t+1} < D_t$ (growing deviations).
- **Expansive:** $D_{t+1} > D_t$ (suppressing deviations).
- **Neutral:** $D_{t+1} \approx D_t$ in expectation (persisting deviations).

Contractive behaviour would eliminate deviations, preventing persistence, while expansive behaviour would amplify them in a way inconsistent with the empirical stability of modern transformers/LLMs (Xiong et al., 2020). Neutral behaviour is therefore the structural condition under which predictive deviations can persist through decoding. Furthermore, as seen in Section 2.2.1, two continuations may converge toward the same correct answer in meaning while their next-step predictive distributions $p_{t+1}$ and $q_{t+1}$ still differ. Hence, neutrality is an architectural *necessary, but not sufficient* condition for the persistence of any deviation, including (semantic) hallucinations.

## 3   NEUTRALITY, PREDICTABLE DRIFT, AND INFERENCE

This section analyzes how predictive deviation evolves from one decoding step to the next. We compare paired autoregressive continuations under the closed an open regimes.

A *rollout* is a full autoregressive continuation, consisting of a sequence of hidden states, decoded distributions, and token draws. A *paired rollout* consists of two continuations that start from the same prompt and evolve under the same model. At step $t$ they have hidden states $(h_t, \tilde{h}_t)$ and predictive distributions $(p_t, q_t)$. Predictive separation is measured by $D_t = \text{JS}(p_t, q_t)$, and the associated one-step change in predictive deviation is the *drift increment*

$$X_t = D_{t+1} - D_t.$$

Since JS divergence is bounded (Lemma 3), $|X_t| \leq \log 2$. To study the evolution over multiple steps we also consider the cumulative drift, which forms the sequence $S_N = \sum_{t=1}^{N} X_t, \quad N \geq 1$.

### 3.1 CLOSED AND OPEN DECODING REGIMES

The drift increment $X_t$ depends on two factors: how the residual architecture transforms the hidden states, and how stochastic differences arise from drawing different tokens during autoregressive sampling. To separate these two sources we use controlled comparisons (open versus closed). In the closed regime both continuations consume the same tokens, removing sampling variability and isolating the architectural update. In the open regime they sample independently, matching natural autoregressive decoding. Both regimes use the same autoregressive transformer and differ only in how the next token is chosen.

**Closed decoding.** In this regime, both continuations consume the same next token $\tau_t \sim p_t$. This removes stochastic branching: the only source of change in predictive separation is the architecture itself. The corresponding increment is $X_t^{\text{closed}} = D_{t+1}^{\text{closed}} - D_t$, which gives the architectural baseline. Appendix A.3 (Lemma 5) shows that in this regime the conditional expectation is zero,

$$\mu_t^{\text{closed}} = \mathbb{E}[X_t^{\text{closed}} \mid \mathcal{F}_t] = 0, \tag{1}$$

which we refer to as *closed neutrality*. It formalizes the idea that pre-LN residual transformers neither contract nor expand predictive differences when no sampling mismatch is introduced.

**Open decoding.** Here, each continuation samples its next token independently: $\tau_t \sim p_t, \quad \tilde{\tau}_t \sim q_t$. This introduces stochastic branching in addition to the architectural update. The corresponding increment is $X_t^{\text{open}} = D_{t+1}^{\text{open}} - D_t$. Appendix A.2 (Lemma 4) shows that in this case, the conditional expectation $\mu_t = \mathbb{E}[X_t^{\text{open}} \mid \mathcal{F}_t]$, which we call the *predictable drift*, decomposes as

$$\mu_t = \mathbb{E}[X_t^{\text{closed}} \mid \mathcal{F}_t] + \Delta_t = \mu_t^{\text{closed}} + \Delta_t,$$

where $\Delta_t$ is the systematic effect of token mismatch. By eq 1, this reduces to $\mu_t = \Delta_t$. Thus the predictable part of open drift is entirely due to token mismatch arising from independent sampling.

### 3.2 THE CONTROLLED RANDOMIZATION NETWORK (CRN)

To obtain unbiased and architecturally faithful measurements of drift we use the controlled randomization network (CRN) defined in Appendix A.4. The CRN evolves three coupled continuations, or *arms*, all sharing the same non-token randomness. Each arm is a full continuation of the same prompt and model, differing only in a prescribed perturbation. The baseline, positive and negative arms are mathematical mirror-image modifications of one another.

At each decoding step the CRN records the JS divergences $D_t$, $D_t^+$ and $D_t^-$ and forms the antisymmetric increment

$$X_t = \frac{1}{2}\Big[(D_{t+1}^+ - D_t^+) - (D_{t+1}^- - D_t^-)\Big].$$

The conditional expectation of the drift increment, $\mu_t = \mathbb{E}[X_t \mid \mathcal{F}_t]$, equals zero in the closed regime (architectural neutrality) and quantifies the predictable effect of token mismatch in the open regime.

The CRN has two structural features. First, the $(+)$ and $(-)$ arms are mathematical mirror images under the same non-token randomness, such that each step compares sign-reversed versions of the same continuation. Second, the conditional expectation of the drift increment, $\mu_t = \mathbb{E}[X_t \mid \mathcal{F}_t]$, separates the closed-regime contribution from the additional effect caused by sampling different tokens, written as $\Delta_t^{\pm}$. The symmetry condition $\Delta_t^+ = \Delta_t^-$ holds when this sampling effect is the same for both mirror arms. In that case $\mu_t = 0$, making architectural neutrality experimentally verifiable rather than purely theoretical. All details and proofs are given in Appendix A.3 (Lemma 5).

Finally, to estimate the CRN conditional mean, we use *sibling rollouts*: independent repeats of the same CRN obtained by resampling only the random seeds. Siblings give repeated evaluations of $X_t$ under the same state $\mathcal{F}_t$. Their average estimates $\mathbb{E}[X_t \mid \mathcal{F}_t]$, which is zero in the closed regime. To separate this systematic component from sampling variability we write

$$X_t = \mu_t + Y_t, \qquad \mu_t := \mathbb{E}[X_t \mid \mathcal{F}_t], \qquad Y_t := X_t - \mu_t.$$

The term $\mu_t$ is fixed once the current hidden state and token distributions are fixed, while $Y_t$ contains the remaining randomness (martingale fluctuations). In the next section we bound $\mu_t$, since only this systematic part is determined by the architecture and can be controlled.

### 3.3 PREDICTABLE DRIFT, DRIFT IDENTITY, AND THE DRIFT CORRIDOR

For paired continuations, the one-step change in predictive divergence is $X_t = D_{t+1} - D_t$. The conditional mean is given by the drift identity (Appendix D, Lemma 13):

$$\mu_t = \mathbb{E}_{i \sim p_t, \, j \sim q_t}\big[D_{t+1}(i,j) - D_{t+1}(i,i)\big], \tag{2}$$

where $i$ and $j$ are the next tokens drawn from $p_t$ and $q_t$. The inner term compares two possible next steps, same token versus different tokens. Taking the expectation over the two next token draws gives the exact drift. Thus $\mu_t$ captures the deterministic effect of token mismatch. The drift identity shows that the value of $\mu_t$ depends only on how much the next-step update changes when the token changes, so obtaining a bound for $\mu_t$ requires a bound on the sensitivity of this update to its token input.

### 3.4 PREDICTABLE DRIFT CORRIDOR

As discussed previously, bounding $\mu_t$ reduces to bounding how sensitive the one-step update is to the token that enters it. The update map $\widetilde{\Phi}_t(\cdot)$ is Lipschitz in the token embeddings. If $E_i$ and $E_j$ are the embeddings of tokens $i$ and $j$, then the one-step update $\widetilde{\Phi}_t(\cdot)$ satisfies

$$\|\widetilde{\Phi}_t(j) - \widetilde{\Phi}_t(i)\|_2 \leq L_{\mathrm{ker},t} \|E_j - E_i\|_2. \tag{3}$$

Combining eq. 3 with the Lipschitz bounds for JS divergence, softmax, and the decoder yields a deterministic interval. We refer to this interval as the *predictable drift corridor*, which implies $\mu_t \in [-c_t, \, c_t]$ and is formalized below:

**Proposition 1** (Predictable drift corridor). *For each step $t$,*

$$|\mu_t| \leq L_{\mathrm{JS},t} \, L_{\mathrm{sm},t} \, \|W\|_2 \, L_{\mathrm{ker},t} \, \mathbb{E}_{i,j} \|E_j - E_i\|_2 =: c_t, \tag{4}$$

*with $E$ the embedding matrix and $E_i$ its token vectors. If decoder matrix $W$ has $\sigma_{\min}(W) > 0$, then*

$$|\mu_t| \leq L_{\mathrm{JS},t} \, L_{\mathrm{sm},t} \, \kappa_2(W) \, L_{\mathrm{ker},t} \, \mathbb{E}_{i,j} \|M_j - M_i\|_2, \tag{5}$$

*where $\kappa_2(W) = \|W\|_2 / \sigma_{\min}(W)$ and $M = WE$ denotes the logit embeddings.*

*Proof sketch.* Lemma 8 in Appendix B bounds the JS change in equation 3 by $L_{\mathrm{JS},t} \|\widetilde{\Phi}_t(j) - \widetilde{\Phi}_t(i)\|_2$. Lemma 9 in Appendix B bounds the one-step update by $L_{\mathrm{ker},t} \|W\|_2 L_{\mathrm{sm},t} \|E_j - E_i\|_2$, which gives equation 4. If $\sigma_{\min}(W) > 0$, distances may be measured in logit space via $M = WE$, yielding equation 5. Full derivations appear in Appendix D, Theorem 6. $\qquad\square$

Consequently, the corridor specifies the maximal deterministic magnitude of predictable drift allowed by the architecture. It does not fix the direction of drift but guarantees that even under open sampling, the systematic effect of token mismatch is bounded.

### 3.5 MARTINGALE STRUCTURE, WE HAVE THE DECOMPOSITION AND CUMULATIVE DRIFT

Recall that we can write $X_t = \mu_t + Y_t$, where $Y_t$ contains the martingale fluctuations. This decomposition induces cumulative predictable and centered components:

$$B_N = \sum_{t \leq N} \mu_t, \qquad M_N = \sum_{t \leq N} Y_t, \qquad S_N = B_N + M_N.$$

A full inference rule must account for both terms. The corridor bounds $c_t$ control the predictable contribution $B_N$, while deviation inequalities control the centered term $M_N$. Neither component is sufficient on its own, as $c_t$ does not describe fluctuations from sampling, and the deviation bounds do not restrict the predictable part. Hence, the two components require different analytical tools, combined in the blended neutrality theorem:

**Theorem 1** (Blended neutrality reporting). *Let $\bar{X}_N = \frac{1}{N} \sum_{t=1}^{N} X_t^{\mathrm{open}}$ and $c_t$ as above. Then*

$$\big|\mathbb{E}[\bar{X}_N]\big| \leq \min\left\{ \frac{1}{N} \sum_{t=1}^{N} c_t, \; \big|\bar{X}_N - \frac{1}{N} \sum_{t=1}^{N} \mu_t\big| + z_{0.975} \frac{\widehat{s}_N}{\sqrt{N}} \right\}, \tag{6}$$

*with $\widehat{s}_N^2 = \frac{1}{N} \sum_{t=1}^{N} (X_t^{\mathrm{open}} - \bar{X}_N)^2$. If $\frac{1}{N} \sum c_t \to 0$, then $\frac{1}{N} \sum \mu_t \to 0$ and the standard error band applies directly to $\bar{X}_N$.*

*Proof sketch.* The deterministic control bounds $\mathbb{E}[\bar{X}_N]$ by $\frac{1}{N} \sum c_t$ (Lemma 10). Boundedness of JS divergence ensures that every increment satisfies $|X_t| \leq \log 2$, and therefore $Y_t = X_t - \mu_t$ is also uniformly bounded (Appendix D). Freedman's inequality provides finite-sample control of the cumulative fluctuation $M_N = \sum_{t \leq N} Y_t$, and the martingale central limit theorem describes its asymptotic behaviour, which gives the Gaussian limit with standard error (Theorem 3 and Theorem 4). The combined statement leads to Theorem 5 in Appendix C. $\square$

The blended reporting rule combines these controls into a finite-sample criterion. It makes neutrality testable in paired rollouts: if the observed drift lies within both bounds, neutrality cannot be rejected.

## 3.6 Agents and the mean field lift

The drift framework developed so far treats each one-step increment $X_t$ as arising from a single paired rollout. To understand neutrality beyond a single rollout, we can interpret $X_t$ as many small agents whose empirical average, or *mean-field*, forms the observed drift. Mean-field models originate in stochastic finance and control theory (Lasry & Lions, 2007; Huang et al., 2006; Yang et al., 2017; Carmona & Delarue, 2018), where large populations of interacting agents are approximated by their empirical distribution. The key principle is that when agents are exchangeable and individually negligible, the empirical law of their actions converges to a deterministic population law. This provides the bridge from finite-sample neutrality to structural neutrality at scale.

### 3.6.1 Agents

We formalize the agent model by defining an *agent* to be one elementary contribution to drift at a fixed step $t$ (Appendix E). We use two agent views:

1. *Trajectory agents*: token pairs $(i, j)$ sampled from $(p_t, q_t)$, each producing one increment $X_{t,a}^{\text{open}}$.
2. *Layerwise agents*: residual blocks $H_\ell$ contributing finite difference drifts across depth.

Agents are exchangeable: their joint law is invariant under permutations (Appendix E, Definition 4). Furthermore, neutrality at the agent level is immediate from the drift identity and closed-regime neutrality: for each agent $a$, $\mathbb{E}[X_{t,a}^{\text{open}} \mid \mathcal{F}_t] = 0$. Because each $X_{t,a}$ is bounded, and the empirical mean of $M$ agents satisfies a conditional law of large numbers, as stated in the following proposition:

**Proposition 2** (Finite-agent neutrality)**.** *For $M$ exchangeable agents,*

$$\bar{X}_t^{(M)} = \frac{1}{M} \sum_{a=1}^{M} X_{t,a} \xrightarrow{\text{a.s.}} 0 \qquad (M \to \infty).$$

*Proof.* Linearity of conditional expectation gives $\mathbb{E}\left[\bar{X}_t^{(M)} \mid \mathcal{F}_t\right] = \frac{1}{M} \sum_{a=1}^{M} \mathbb{E}[X_{t,a} \mid \mathcal{F}_t] = 0$. $\square$

### 3.6.2 Population limit

In the mean-field limit, where the number of agents $M$ tends to infinity, the empirical distribution of agent actions converges to a deterministic law. Because each finite-agent system is neutral, the limit inherits neutrality, which is formalized below:

**Theorem 2** (Mean-field neutrality)**.** *Building on Proposition 2, assume the agent actions $\{X_t^{(a)}\}_{a=1}^{M}$ are exchangeable and satisfy $\mathbb{E}[X_t^{(a)} \mid \mathcal{F}_t] = 0$ with bounded second moment. Then*

$$\bar{X}_t^{(M)} = \frac{1}{M} \sum_{a=1}^{M} X_t^{(a)} \xrightarrow[M \to \infty]{a.s.} 0,$$

*so the population law inherits neutrality. Moreover, the predictable corridor $c_t$ and the blended reporting rule (Theorem 1) extend to the mean-field limit without modification.*

*Proof sketch.* By Proposition 2, each finite-agent system has $\mathbb{E}[\bar{X}_t^{(M)} \mid \mathcal{F}_t] = 0$. Exchangeability then allows a de Finetti representation, and the law of large numbers for exchangeable sequences implies $\bar{X}_t^{(M)} \to 0$ almost surely as $M \to \infty$. Bounded increments guarantee that the martingale

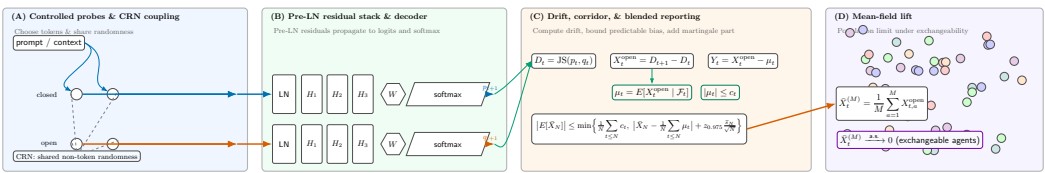

Figure 1: Neutrality audit framework. (A) Closed versus open regime. (B) Residual stack propagates hidden states into token distributions. (C) Drift decomposition: closed increments (martingale differences), open increments ($\mu_t$ bounded by $c_t$ and a centered martingale part). (D) Mean-field lift: neutrality aggregates to the population level.

concentration and corridor bounds apply uniformly, so both carry over to the mean-field limit. See Appendix E, Theorem 7 for the full proof. □

Viewing drift contributions as agents makes two points explicit. First, neutrality is not limited to a single paired continuation but remains present when many contributors are combined. Second, the predictable corridor and the blended reporting rule have the same interpretation at population scale, because the architecture still bounds the deterministic part of drift and the random part continues to satisfy the same martingale controls.

## 4  EXPERIMENTS

In this section, we empirically test the neutrality predictions of Section 3 using the open and closed probes defined in Section 3.1. For each model and each of $K = 32$ prompts with three master seeds, we generate a two-arm CRN pair, expand each arm into $M = 16$ siblings, and decode for $N = 32$ steps at temperature $T = 1.0$. At every step we compute $X_t = D_{t+1} - D_t$. Pooling over prompts, seeds, and siblings yields one closed and one open increment sample per model.

We evaluate these predictions across four GPT2 models (`sshleifer/tiny-gpt2`, `distilgpt2`, `gpt2-medium`, `gpt2-large`), reporting results at both the trajectory and layer-level.

**Hypotheses.** We define the following hypotheses, consistent with the open and closed regime:

$$\text{H1 (closed): } \mathbb{E}[X_t^{\text{closed}} \mid \mathcal{F}_t] = 0, \qquad \text{H2 (open): } |\mu_t^{\text{open}}| \leq c_t.$$

**Tests and controls.** H1 is assessed by a one-sample mean t-test on the pooled closed increments and by Azuma Hoeffding bands for the cumulative sum. H2 is assessed by estimating $\mu_t$ from open increments and checking that all values remain within corridor bounds. The blended neutrality rule (Theorem 1) provides the joint criterion we use in interpretation: open increments must remain within the deterministic corridor $c_t$, and their fluctuations must agree with the martingale behaviour quantified by the Azuma–Hoeffding and anytime $e$-process bounds. Placebo and label randomized CRN pairs serve as controls and should show no systematic drift.

**Population summaries.** Trajectory level summaries use the exchangeability structure required for the mean-field lift in Section 3.6.2. We report the pooled trajectory mean and its confidence bands. Block-level bootstrap intervals are included only as an internal diagnostic.

**Robustness.** The audit is repeated for temperatures $\{0.5, 1.0, 1.7, 2, 5\}$ and sibling counts $\{4, 8, 16\}$, with the same protocol applied to all models. These ablation studies are reported in Appendix F.

### 4.1  MAIN RESULTS

As our results confirm, closed probes behave as bounded martingale differences. The increments are centered, yet as martingales they wander, which produces variability on the scale of the square root of the horizon. In Figure 2, this behavior is visible in the wide spread of trajectories for `gpt2-large`. The black mean line remains centered while individual paths fluctuate within the Azuma–Hoeffding envelope. Table 1 confirms that the mean drift is close to zero, the $t$-tests across prompts are non-significant with $p \geq 0.14$, and the Azuma coverage is complete. These results validate the theoretical

claim that closed probes should show neutrality in expectation while still displaying substantial pathwise variance.

Open probes differ in that their increments may contain predictable drift, bounded in theory by the corridor. Figure 2 shows that this drift is numerically tiny. The mean path remains flat and the grey confidence ribbon collapses around zero, even though some individual trajectories diverge as expected when tokens are decoupled. The aggregate results in Table 1 show mean drifts of order $10^{-8}$ to $10^{-10}$, with all values well inside the corridor. The only marginal case appears for `distilgpt2`, where the prompt-level $t$-test reports $p = 4.46 \times 10^{-2}$. This effect disappears under the anytime $e$-test, which returns $E_{\max} = 1.000$ and $p_e = 0.906$, indicating no sustained deviation from neutrality.

Across model scales the evidence is consistent. The smallest variant, `tiny`, and the largest, `large`, both satisfy the neutrality predictions in closed and open configurations. The intermediate models `distil` and `medium` follow the same pattern. Table 1 shows that mean drifts remain negligible, $t$-tests do not reject, and $e$-test maxima remain close to one across all cases. This stability across scale indicates that neutrality is a structural feature of the GPT-2 residual architecture rather than a property that depends on parameter count.

Layer-level diagnostics add another perspective. When each residual block is treated as an agent, the estimates of drift remain centered near zero with confidence intervals that cover both positive and negative values. Table 6 shows this explicitly for `tiny` and `distil`. The intervals are wide, which reflects the limited sample size at this granularity, but the absence of systematic deviation suggests that no individual block introduces consistent bias.

| Model | Params (M) | Probe | Mean drift | 95% CI | $t$-test $p$ | Azuma coverage | Emax / $p_e$ |
|---|---|---|---|---|---|---|---|
| `tiny-gpt2` | 15 | Closed | $3.022e{-}11$ | $[-1.877e{-}10,\ 3.468e{-}07]$ | $9.980e{-}01$ | 1/1 (100%) | 1.000 / 0.794 |
| | | Open | $-1.281e{-}11$ | – | $9.990e{-}01$ | – | 1.000 / 1.000 |
| `distilgpt2` | 82 | Closed | $1.385e{-}04$ | $[-1.839e{-}02,\ 1.843e{-}02]$ | $9.700e{-}01$ | 5/5 (100%) | 1.117 / 0.969 |
| | | Open | $-1.496e{-}08$ | – | $3.900e{-}01$ | – | 1.000 / 1.000 |
| `gpt2-medium` | 345 | Closed | $-8.917e{-}04$ | $[-1.687e{-}02,\ 1.630e{-}02]$ | $1.560e{-}01$ | 174/174 (100%) | 1.049 / 0.148 |
| | | Open | $1.477e{-}09$ | – | $4.710e{-}01$ | – | 1.000 / 1.000 |
| `gpt2-large` | 774 | Closed | $6.467e{-}06$ | $[-2.851e{-}02,\ 2.788e{-}02]$ | $9.860e{-}01$ | 193/193 (100%) | 1.792 / 0.982 |
| | | Open | $-6.038e{-}10$ | – | $3.240e{-}01$ | – | 1.000 / 1.000 |

Table 1: Trajectory level neutrality audit results for four GPT2 model scales. The table95% CIs, prompt level $t$ tests, Azuma coverage for closed trajectories, and anytime $e$ test statistics for both closed and open probes.

| Model | Block | $\hat{\mu}$ | SE | 95% CI |
|---|---|---|---|---|
| `tiny-gpt2` | All (4) | $1.157e{-}07$ | $9.729e{-}08$ | $(-1.877e{-}10,\ 3.468e{-}07)$ |
| `distilgpt2` | All (4) | $-2.833e{-}04$ | $9.159e{-}03$ | $(-1.839e{-}02,\ 1.843e{-}02)$ |
| `gpt2-medium` | All (4) | $-2.844e{-}04$ | $8.178e{-}03$ | $(-1.687e{-}02,\ 1.630e{-}02)$ |
| `gpt2-large` | All (4) | $1.221e{-}04$ | $1.341e{-}02$ | $(-2.851e{-}02,\ 2.788e{-}02)$ |

Table 2: Layer-as-agent diagnostics for prompt 1. Reported are the mean action $\hat{\mu}$, its standard error, and a 95% CI aggregated across residual blocks. Entries marked (–) indicate that estimates were not computed for that model in this run.

For `gpt2-large`, the contrast between closed and open probes is clear in Figure 2. Closed probes force both trajectories to consume the same tokens, so increments are martingale differences with $\mathbb{E}[X_t^{\text{closed}}|\mathcal{F}_t] = 0$. As martingales, they wander: over $N = 32$ steps the cumulative drift fluctuates on the $\sqrt{N}$ scale, and Azuma–Hoeffding only guarantees a loose pathwise envelope. Nevertheless, every trajectory remains inside this envelope, so the wide fluctuations seen in the closed panel are consistent with neutrality and not evidence of bias.

Open probes decouple token draws, introducing a predictable drift $\mu_t$. Theory bounds this drift by the corridor $c_t$, and in practice, it is much smaller than the variance of closed wandering. Because we average over $K = 32$ prompts, three seeds, and $M = 16$ siblings (more than 1500 paths per step), these small biases nearly cancel. The result is that the mean cumulative drift is extremely stable, with a confidence interval on the order of $10^{-7}$ that visually collapses onto the black curve. Individual open trajectories may diverge, as expected, but what matters is that the mean remains neutral within the predictable corridor. This pattern matches the predictions of Section 3: closed probes reveal

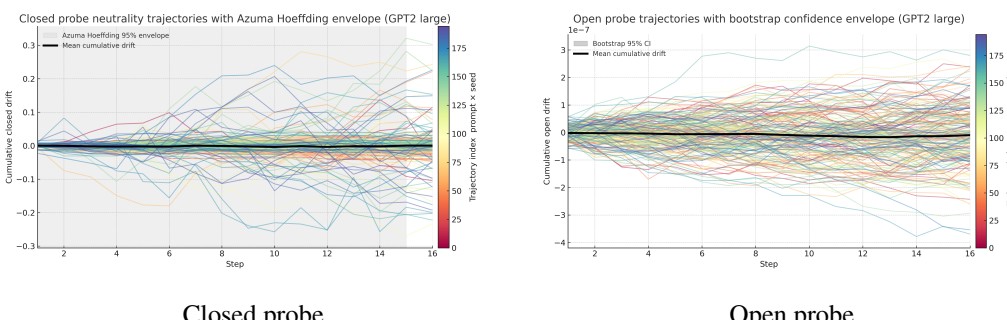

Closed probe.                    Open probe.

Figure 2: Neutrality audits for `gpt2-large`. Closed probes remain within the Azuma–Hoeffding envelope, while open probes yield an extremely stable mean drift whose confidence band is visually indistinguishable from the curve.

martingale fluctuations, while open probes confirm that structural drift is negligible once averaged. Ablation studies varying $T$ and $M$ (Appendix F) further confirm the neutrality results.

## 4.2 SCALING PREDICTIVE NEUTRALITY BEYOND GPT-2

The neutrality analysis does not rely on properties specific to the GPT-2 family. It uses only the residual update rule shared across modern pre-LN transformers. The audit of Qwen2.5 models spanning 0.5B to 3B parameters in Table 3 demonstrates this directly. Although Qwen is trained with a separate data pipeline and training procedure, it reproduces all neutrality signatures: closed probe drifts remain statistically indistinguishable from zero under the prompt level t–test, every trajectory stays within its Azuma envelope, and the anytime e–test shows no systematic growth. The open regime exhibits the same pattern.

| Model | Params (M) | Probe | Mean drift | 95% CI | $t$ test $p$ | Azuma coverage | Emax / $p_e$ |
|---|---|---|---|---|---|---|---|
| Qwen2.5-0.5B-Instruct | 500 | Closed | $-2.611e-03$ | $[-6.813e-03, 1.591e-03]$ | $2.040e-01$ | 45/45 (100%) | 1.000 / 0.173 |
| | | Open | $-3.792e-04$ | $[-6.282e-03, 5.523e-03]$ | $8.924e-01$ | – | 1.176 / 0.867 |
| Qwen2.5-1.5B-Instruct | 1500 | Closed | $-1.102e-03$ | $[-4.366e-03, 2.161e-03]$ | $4.807e-01$ | 45/45 (100%) | 1.741 / 0.572 |
| | | Open | $-2.321e-04$ | $[-4.438e-03, 3.974e-03]$ | $9.075e-01$ | – | 1.013 / 0.907 |
| Qwen2.5-3B-Instruct | 3000 | Closed | $-9.170e-04$ | $[-4.611e-03, 2.777e-03]$ | $6.028e-01$ | 45/45 (100%) | 1.014 / 0.544 |
| | | Open | $-5.840e-04$ | $[-3.786e-03, 2.618e-03]$ | $7.015e-01$ | – | 1.290 / 0.801 |

Table 3: Trajectory level neutrality audit results for Qwen2.5 0.5-3B models using 3 seeds and 15 prompts. The table reports mean drift estimates, 95% CIs, prompt level $t$ tests, Azuma coverage for closed trajectories, and anytime $e$ test statistics for both closed and open probes.

**Evaluation of Hypotheses** For the closed probes, Figure 2 shows centered but wandering paths, and Table 1 and Table 3 confirm that drifts stay near zero with full Azuma coverage and flat $e$-process. Hypothesis 1 holds: increments act as martingale differences, neutral in expectation. For the open probes, trajectories may diverge, yet their mean drift is small. The results report values below the corridor constants and non-rejections under $e$-tests. Hypothesis 2 holds: predictable drift remains bounded and negligible across scales.

## 5 DISCUSSION

**Neutrality.** Our experiments confirm the neutrality properties proven in Section 3. Closed probes behave as bounded martingales: they show no systematic drift. Open probes admit predictable drift, but the observed values remain several orders of magnitude below corridor bounds. Together the results in Figures 2 and Tables 1–3 establish that the residual architecture neither contracts nor expands deviations in expectation.

**Mean–field dynamics.** The mean–field formulation explains why these results scale. At the trajectory level, increments behave as martingale differences, and under exchangeability this neutrality law lifts to the population of token–agents. At the layer level, residual blocks act as agents through their finite–difference contributions, and bootstrap intervals show that these actions fluctuate but remain

centered. In both cases the collective dynamics are neutral rather than adversarial, so the population equilibrium is persistence. The novelty is that neutrality is not confined to single increments but survives the mean–field lift, turning a local property into a system–level invariant.

**Hallucination persistence.** The central question in this paper was what structural rule governs how two continuations evolve once a deviation is already present. The experiments give a consistent answer. The closed probes show that predictive differences wander without a restoring force, and the open probes show that this wandering is not driven by systematic expansion because the predictable drift remains inside the corridor. The same pattern appears across all GPT2 scales, and the block summaries do not reveal any consistent source of bias. At the trajectory-level a similar conclusion holds for the Qwen 2.5 model. These findings describe predictive behaviour only. Neutrality is necessary, but not sufficient, for semantic hallucinations to persist after onset. It is necessary because it allows predictive differences to continue rather than collapse, yet not sufficient because a model can still correct the meaning even when the predictive distributions differ. The results also shed light on why LLMs often appear to have a short memory. If predictive differences are allowed to persist without a contracting pull, then earlier context cannot reliably influence later tokens, and small mismatches can remain visible even when the output appears fluent.

**Implications.** The consequence is that hallucination persistence is an architectural invariant rather than a byproduct of training. Approaches that control onset, such as entropy reduction, retrieval augmentation, or reinforcement learning with human feedback, cannot by themselves eliminate persistence, since the backbone dynamics remain neutral once a deviation has occurred. Mitigation therefore requires structural interventions: architectures that introduce contraction or external anchoring mechanisms that continuously re-ground the generation. By combining statistical probes with a mean–field lift, we provide a non-anthropomorphic language to describe this mechanism, framing hallucinations as a structural feature of residual transformers that persists across scales.

## 5.1 LIMITATIONS

Our empirical scope is limited by the requirement that models expose full control over sampling. GPT-2 variants satisfy this constraint, and the audit of Qwen2.5 –0.5-3B shows that neutrality extends beyond the GPT-2 family to an independently trained, mid scale modern architecture. The main limitations are practical: larger models require many more trajectories and seeds, and proprietary systems do not expose the interfaces needed for controlled probes.

The horizon was set to $N = 32$ for computational reasons. The neutrality theorem is time uniform and does not depend on $N$, and this length already shows neutral wandering and bounded drift. It is however, too short to study very long generations, so larger $N$ would provide an additional empirical check without affecting the theoretical claim.

At the architectural level, the trajectory results follow from formal martingale arguments. The mean-field lift uses exchangeability of paired increments along a trajectory. The layer-level diagnostic inspects individual blocks, which are not exchangeable because they have different positions, functions, and scales. Bootstrap intervals therefore give only a rough internal check. The audit isolates architectural neutrality. It does not examine how this interacts with training procedures such as reinforcement learning from human feedback or retrieval augmented generation.

## 6 CONCLUSION

We showed that persistence follows from a simple architectural fact: pre-LayerNorm residual transformers do not pull paired rollouts together or push them apart in expectation. The stepwise drift identity makes this explicit, and the operator bounds for LayerNorm, the residual stack, and the decoder give a predictable drift corridor that limits how much systematic separation can occur at each step. The blended reporting rule links this structural limit to finite sample estimates, providing a direct test for neutrality. A mean field lift then shows that the same neutral behaviour appears when many local drift contributions are combined, which explains the stability of the effect across prompts and model scales. Empirically, GPT2 and Qwen 2.5 audits align with these predictions. Interventions that do not modify the residual backbone can curb onset but cannot eliminate persistence once deviations arise. In this sense, trajectories may wander, but under neutrality their drift remains bounded and unbiased—reminding us that not all who wander are lost (Tolkien, 1954).

REPRODUCIBILITY STATEMENT

The experimental protocol is defined in Section 4, which specifies the closed and open probes, the controlled–randomization coupling, the decoding horizon, sibling structure, and the statistical tests used in the neutrality audit. The accompanying Colab scripts implements this protocol and is adapted to support a broader set of open models beyond GPT–2.

**Entry point**    The main results of the paper are reproduced with:

`neutrality_audit.py`

This script runs the full neutrality protocol. It produces `results.csv` and `msgs.csv` together with the ablation outputs reported in Appendix F.

For completeness, we also provide a second script,

`model_agnostic_neutrality_audit.py,`

which offers a modular model-independent implementation of the closed and open probes. This lighter script is used for the Qwen 2.5 runs in Section 4. It omits the layer-as-agent diagnostic and the ablation sweeps because of the costly computations.

**Models.**    All GPT–2 models used in the paper are loaded directly from HuggingFace without modification. The script generalizes automatically to any HuggingFace `CausalLM` model with standard cache semantics, including Qwen and Llama.

For Qwen, the scaling experiments with 1.5 and 3 billion parameters, load the weights in `float16` rather than `float32` because of model size constraints. Although Qwen is loaded in `float16`, the neutrality audit uses matched fp16 forward passes (baseline, $+\varepsilon$, $-\varepsilon$), which cancels first order quantisation effects. The remaining fp16 noise is far smaller than the empirical variability across prompts and seeds, so the drift estimates and statistical tests are unaffected.

Evaluating Llama models requires accepting Meta's model license on a HuggingFace account before access is granted. Once downloaded, the same probe definitions and statistical tests apply unchanged. Larger models may require a reduced decoding horizon or prompt count for runtime feasibility.

**Prompts and configuration.**    Prompts are defined explicitly inside the scripts as a Python list. The scripts expose all core parameters (number of steps, siblings, temperature, seeds, and prompts) at the top. Setting them to the values in Section 4 yields the behaviour reported in the main tables. The scripts include a lightweight pilot run that estimates variance and required sample size under a small closed probe configuration. This pilot is used only for compute management and debugging when resources are limited, and it is not part of the theoretical protocol or the reported results.

The Qwen experiments use a fixed set of 3 seeds and 15 prompts for computational management purposes.

**Randomness control.**    NumPy and PyTorch RNGs are seeded deterministically. Closed probes reuse the same token draws across both arms, implementing the CRN requirement. Open probes use independent draws. Re–running with the same master seed reproduces the printed statistics.

**Dependencies and hardware.**    The scripts depends on PyTorch, Transformers, NumPy, and SciPy. It runs on standard Colab GPUs (e.g. T4, A100), with CPU fallback supported for smaller models. No proprietary APIs or external services are required. All models are loaded through the HuggingFace hub. Open models require no additional services, while Llama weights require accepting Meta's license before download.

STATEMENT USE OF LLM

Large language models were used for polishing language, fixing minor coding errors, and triaging related work. The proofs, analyses, and results were developed by the authors, and all cited references, including linked sources when available, were manually verified.

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

## APPENDIX CONTENTS

## SYMBOLS AND NOTATION

| Symbol | Meaning |
| --- | --- |
| $V$ | Vocabulary size. Index set $[V] = \{1, \dots, V\}$. |
| $\Delta^{V-1}$ | Probability simplex over $V$ tokens. |
| $\mu(x), \sigma(x)$ | Coordinate mean and standard deviation of $x$ used by LayerNorm. |
| $\varepsilon$ | LayerNorm stabilizer, strictly positive. |
| $\gamma, \beta$ | LayerNorm gain and bias, with $\|\gamma\|_\infty = \max_i |\gamma_i|$. |
| LN | LayerNorm map $\gamma \odot (x - \mu(x)\mathbf{1})/\sigma(x) + \beta$. |
| $J_{\mathrm{LN}}(x)$ | Jacobian of LayerNorm at $x$. |
| $S$ or $s_T$ | Softmax map. With temperature $T$, $s_T(z)_i = \exp(z_i/T)/\sum_j \exp(z_j/T)$. |
| $T$ | Sampling temperature controlling softmax sharpness and entropy. |
| $z \in \mathbb{R}^V$ | Logit vector. |
| $p, q \in \Delta^{V-1}$ | Paired next token distributions. |
| $\mathrm{KL}(p\|q)$ | Kullback Leibler divergence. |
| $\mathrm{JS}(p, q)$ | Jensen Shannon divergence bounded by $\log 2$. |
| $m$ | Mixture $m = \frac{1}{2}(p + q)$ used in JS definitions. |
| $t$ | Decoding step index. |
| $h_t, \tilde{h}_t$ | Paired hidden states at step $t$ for the two arms. |
| $p_t, q_t$ | Paired decoded next token distributions at step $t$. |
| $D_t$ | Predictive divergence at step $t$, $D_t = \mathrm{JS}(p_t, q_t)$. |
| $\mathcal{F}_t$ | Filtration up to step $t$ generated by states, tokens, and couplings. |
| $\xi_{t+1}$ | Exogenous randomness for the one step transition, shared across arms. |
| $\mathcal{G}_t$ | Enlarged sigma algebra $\sigma(\mathcal{F}_t, \xi_{t+1})$. |
| $\tau_t, \tilde{\tau}_t$ | Tokens consumed at step $t$ by the two arms. |
| $K(h_t, i; \xi_{t+1})$ | One step kernel mapping token $i$ to the next hidden state. |
| $D_{t+1}(i, j; \xi_{t+1})$ | Next step divergence if arm one takes $i$ and arm two takes $j$. |
| $X_t^{\mathrm{closed}}, X_t^{\mathrm{open}}$ | Closed and open increment. |
| $\mu_t$ | Predictable drift $\mathbb{E}[X_t^{\mathrm{open}} \mid \mathcal{F}_t]$. |
| $\Delta_t$ | Token mismatch drift term defined in Lemma 4. |
| $D_t^{\pm}$ | Divergences for the $+$ and $-$ CRN arms. |
| $X_t$ | CRN antisymmetric increment. |
| $X_{t,\mathrm{closed}}^{\pm}$ | Closed increments in each CRN arm. |
| $M$ | Sibling count used for variance reduction. |
| $X_t^{(m)}, \overline{X}_t$ | Increment from sibling $m \in \{1, \dots, M\}$ and sibling average. |
| $W$ | Decoder matrix mapping hidden state to logits. |
| $\|W\|_2, \sigma_{\min}(W)$ | Spectral norm of $W$, smallest singular value of $W$. |
| $\kappa_2(W)$ | Condition number $\|W\|_2/\sigma_{\min}(W)$. |
| $E$ | Embedding matrix. Column $E_i$ is the embedding of token $i$. |
| $M = WE$ | Logit space embeddings, with columns $M_i$. |
| $\Phi_t(i), \widetilde{\Phi}_t(j)$ | Decoded distributions after one-step kernel updates. |
| $L_{\mathrm{sm},t}$ | Softmax Lipschitz constant at step $t$, bounded by $1/(2T)$. |
| $L_{\mathrm{JS},t}$ | Local Lipschitz constant of JS in its second argument at step $t$. |
| $L_{\mathrm{ker},t}$ | Kernel Lipschitz constant up to step $t$, product of block constants. |
| $H_\ell(x)$ | Residual block $x + G_\ell(\mathrm{LN}(x))$. |
| $L_\ell$ | Lipschitz constant of block $\ell$. |
| $c_t$ | Corridor radius bounding $|\mu_t|$, defined in Proposition 3. |
| $Y_t$ | Centered increment $X_t^{\mathrm{open}} - \mu_t$. |
| $M_N, B_N$ | Martingale sum $\sum_{t=1}^N Y_t$, drift sum $\sum_{t=1}^N \mu_t$. |
| $V_N$ | Predictable quadratic variation $\sum_{t=1}^N \mathbb{E}[Y_t^2 \mid \mathcal{F}_{t-1}]$. |
| $\bar{X}_N$ | Empirical mean drift $\frac{1}{N}\sum_{t=1}^N X_t^{\mathrm{open}}$. |
| $\widehat{s}_N$ | Sample standard deviation of open increments. |
| $z_{0.975}$ | Standard normal quantile for 95 percent two sided bands. |
| $E_{\max}, \alpha$ | Maximum of the anytime $e$ process. Test size, rejection when $E_{\max} \geq 1/\alpha$. |
| $A_k$ | Generic exchangeable agent variable in Appendix E. |

# A    PRELIMINARIES

## A.1    AUTOREGRESSIVE COMPONENTS

**Lemma 1** (LayerNorm operator norm). *Let* $\mathrm{LN} : \mathbb{R}^d \to \mathbb{R}^d$ *be*

$$\mathrm{LN}(x) \;=\; \gamma \odot \frac{x - \mu(x)\mathbf{1}}{\sigma(x)} + \beta, \qquad \mu(x) = \tfrac{1}{d} \sum_{i=1}^{d} x_i, \quad \sigma(x) = \sqrt{\tfrac{1}{d} \sum_{i=1}^{d} (x_i - \mu(x))^2 + \varepsilon},$$

*with* $\varepsilon > 0$ *and* $\|\gamma\|_\infty = \max_i |\gamma_i|$. *Then, for all* $x \in \mathbb{R}^d$,

$$\left\| J_{\mathrm{LN}}(x) \right\|_2 \;\le\; \frac{\|\gamma\|_\infty}{\sqrt{\varepsilon}}.$$

*Proof.* Define $c(x) = x - \mu(x)\mathbf{1}$ and $P = I - \tfrac{1}{d}\mathbf{1}\mathbf{1}^\top$, so that $c(x) = Px$ and $\|P\|_2 = 1$. Then

$$\hat{x} \;=\; \frac{c(x)}{\sigma(x)} \;=\; \frac{Px}{\sigma(x)}, \qquad \mathrm{LN}(x) \;=\; \mathrm{Diag}(\gamma)\,\hat{x} + \beta.$$

Thus $J_{\mathrm{LN}}(x) = \mathrm{Diag}(\gamma)\,J_{\hat{x}}(x)$, and therefore

$$\|J_{\mathrm{LN}}(x)\|_2 \;\le\; \|\,\mathrm{Diag}(\gamma)\|_2\,\|J_{\hat{x}}(x)\|_2 = \|\gamma\|_\infty\,\|J_{\hat{x}}(x)\|_2.$$

It remains to show $\|J_{\hat{x}}(x)\|_2 \le 1/\sigma(x)$. For $v \in \mathbb{R}^d$, using $\mu'(x)[v] = \tfrac{1}{d}\mathbf{1}^\top v$ and

$$\sigma^2(x) = \tfrac{1}{d}\|Px\|_2^2 + \varepsilon, \qquad (\sigma^2)'[v] = \tfrac{2}{d}\,(Px)^\top(Pv), \qquad \sigma'(x)[v] = \frac{(Px)^\top(Pv)}{d\,\sigma(x)},$$

we compute

$$J_{\hat{x}}(x)\,v = \frac{Pv}{\sigma(x)} - \frac{Px}{\sigma(x)^2}\,\sigma'(x)[v] = \frac{1}{\sigma(x)}\Big(I - \frac{Px\,(Px)^\top}{d\,\sigma(x)^2}\Big)Pv.$$

Set

$$u = \frac{Px}{\sqrt{d}\,\sigma(x)},$$

so that $\|u\|_2^2 = \frac{\|Px\|_2^2}{d\sigma(x)^2} = 1 - \frac{\varepsilon}{\sigma(x)^2} \le 1$, and

$$\frac{Px\,(Px)^\top}{d\,\sigma(x)^2} = uu^\top.$$

Hence

$$J_{\hat{x}}(x) \;=\; \frac{1}{\sigma(x)}\,(I - uu^\top)\,P.$$

Now $P$ is an orthogonal projector, so $\|P\|_2 = 1$. The matrix $I - uu^\top$ is symmetric with eigenvalues $1$ on $u^\perp$ and $1 - \|u\|_2^2$ on $\mathrm{span}\{u\}$, all in $[0,1]$. Thus $\|I - uu^\top\|_2 = 1$. Therefore

$$\|J_{\hat{x}}(x)\|_2 \;\le\; \frac{1}{\sigma(x)}\,\|I - uu^\top\|_2\,\|P\|_2 \;\le\; \frac{1}{\sigma(x)}.$$

Combining the estimates gives

$$\|J_{\mathrm{LN}}(x)\|_2 \;\le\; \frac{\|\gamma\|_\infty}{\sigma(x)} \;\le\; \frac{\|\gamma\|_\infty}{\sqrt{\varepsilon}},$$

since $\sigma(x) \ge \sqrt{\varepsilon}$. $\qquad\square$

**Remark.** If $Px \neq 0$ and $\gamma = \gamma_0 \mathbf{1}$, then $J_{\hat{x}}(x)$ acts as $v \mapsto v/\sigma(x)$ on the subspace

$$\{ v \in \mathrm{range}(P) : v \perp Px \},$$

so $\|J_{\mathrm{LN}}(x)\|_2 = \|\gamma\|_\infty/\sigma(x)$ and the scaling in $\sigma(x)$ and $\varepsilon$ is sharp.

**Lemma 2** (Softmax Lipschitz constant). *Let $s_T(z) = \mathrm{softmax}_T(z)$ with temperature $T > 0$, so that $p = s_T(z)$ and*

$$s_T(z)_i = \frac{e^{z_i/T}}{\sum_j e^{z_j/T}}.$$

*Then the Jacobian satisfies, for all $z \in \mathbb{R}^V$,*

$$\left\|\nabla s_T(z)\right\|_2 = \frac{1}{T}\left\|\mathrm{Diag}(p) - pp^\top\right\|_2 \leq \frac{1}{2T}.$$

*Moreover, the constant $1/(2T)$ is tight (attained for $V = 2$, $p = (\frac{1}{2}, \frac{1}{2})$).*

*Proof.* Differentiating directly gives

$$\frac{\partial s_T(z)_i}{\partial z_k} = \frac{1}{T}\left(p_i\,\delta_{ik} - p_ip_k\right),$$

hence

$$\nabla s_T(z) = \tfrac{1}{T}\left(\mathrm{Diag}(p) - pp^\top\right).$$

The matrix $\mathrm{Diag}(p) - pp^\top$ is symmetric positive semidefinite, so

$$\left\|\nabla s_T(z)\right\|_2 = \frac{1}{T}\,\lambda_{\max}\left(\mathrm{Diag}(p) - pp^\top\right).$$

For any unit vector $v \in \mathbb{R}^V$,

$$v^\top\left(\mathrm{Diag}(p) - pp^\top\right)v = \sum_i p_iv_i^2 - \left(\sum_i p_iv_i\right)^2 = \mathrm{Var}_p(v),$$

the variance of the random variable that takes value $v_i$ with probability $p_i$.

Let $\alpha = \min_i v_i$ and $\beta = \max_i v_i$. By Popoviciu's inequality,

$$\mathrm{Var}_p(v) \leq \frac{(\beta - \alpha)^2}{4}.$$

Moreover, by Cauchy–Schwarz,

$$(\beta - \alpha)^2 = (|\beta| + |\alpha|)^2 \leq 2(\beta^2 + \alpha^2) \leq 2\sum_i v_i^2 = 2,$$

since $\|v\|_2 = 1$. Combining gives

$$\mathrm{Var}_p(v) \leq \frac{1}{4}(\beta - \alpha)^2 \leq \frac{1}{2}.$$

Taking the supremum over unit vectors $v$ shows

$$\lambda_{\max}\left(\mathrm{Diag}(p) - pp^\top\right) \leq \tfrac{1}{2}.$$

**Tightness.** For $V = 2$, $p = (\frac{1}{2}, \frac{1}{2})$, the matrix

$$\mathrm{Diag}(p) - pp^\top = \begin{bmatrix} \frac{1}{4} & -\frac{1}{4} \\ -\frac{1}{4} & \frac{1}{4} \end{bmatrix}$$

has eigenvalues $0$ and $1/2$. Thus

$$\|\nabla s_T(z)\|_2 = \frac{1}{T} \cdot \frac{1}{2} = \frac{1}{2T},$$

so the bound is attained. $\qquad\square$

## A.2 Divergences

**Lemma 3** (JS divergence). *For all $p, q \in \Delta^{V-1}$ and natural logarithm,*

$$0 \;\le\; \mathrm{JS}(p,q) \;\le\; \log 2.$$

*Proof.* Nonnegativity follows from convexity of KL and symmetry. For the upper bound, let $m = \frac{1}{2}(p+q)$. By Gibbs' inequality, $\mathrm{KL}(p\|m) \le \log \sum_i \frac{p_i^2}{m_i} \le \log 2$, and similarly for $q$; averaging yields the claim. Standard proofs appear in Endres & Schindelin (2003). □

## A.3 Probe kernels

**Lemma 4** (Open-probe kernel). *Fix $t \ge 0$ and let $\mathcal{F}_t$ be the natural filtration up to step $t$, so $(h_t, \tilde{h}_t, p_t, q_t)$ are $\mathcal{F}_t$–measurable and $D_t = \mathrm{JS}(p_t, q_t)$. Let $\xi_{t+1}$ denote all exogenous random variables used by the one–step kernel at time $t+1$, coupled across the two arms and independent of $(\tau_t, \tilde{\tau}_t)$ given $\mathcal{F}_t$, and set $\mathcal{G}_t := \sigma(\mathcal{F}_t, \xi_{t+1})$. For tokens $(\tau, \tilde{\tau})$ define*

$$D_{t+1}(\tau, \tilde{\tau}; \xi_{t+1}) := \mathrm{JS}\Big(S\big(K(h_t, \tau; \xi_{t+1})\big),\, S\big(K(\tilde{h}_t, \tilde{\tau}; \xi_{t+1})\big)\Big).$$

*Let $X_t^{\mathrm{closed}} = D_{t+1}^{\mathrm{closed}} - D_t$ be the increment when both arms consume the same token $\tau_t \sim p_t$, and $X_t^{\mathrm{open}} = D_{t+1}^{\mathrm{open}} - D_t$ the increment when $\tau_t \sim p_t$ and $\tilde{\tau}_t \sim q_t$ are independent. Then*

$$\mathbb{E}[X_t^{\mathrm{open}} \mid \mathcal{F}_t] \;=\; \mathbb{E}\big[X_t^{\mathrm{closed}} \mid \mathcal{F}_t\big] \;+\; \Delta_t,$$

*where*

$$\Delta_t \;=\; \mathbb{E}[D_{t+1}(\tau_t, \tilde{\tau}_t; \xi_{t+1}) - D_{t+1}(\tau_t, \tau_t; \xi_{t+1}) \mid \mathcal{F}_t].$$

*Proof.* All statements are conditional on $\mathcal{F}_t$. First note that $D_t = \mathrm{JS}(p_t, q_t)$ depends only on $(p_t, q_t)$, hence it is $\mathcal{F}_t$–measurable. Therefore $\mathbb{E}[D_t \mid \mathcal{F}_t] = D_t$ in both probe regimes.

Let $\xi_{t+1}$ denote all exogenous randomness used at time $t+1$, independent of $(\tau_t, \tilde{\tau}_t)$ given $\mathcal{F}_t$ and coupled across both arms, and set $\mathcal{G}_t := \sigma(\mathcal{F}_t, \xi_{t+1})$. For fixed $\xi_{t+1}$, the map

$$(\tau, \tilde{\tau}) \mapsto D_{t+1}(\tau, \tilde{\tau}; \xi_{t+1})$$

is deterministic and measurable.

In the open probe,

$$\mathbb{E}[X_t^{\mathrm{open}} \mid \mathcal{G}_t] = \mathbb{E}[D_{t+1}(\tau_t, \tilde{\tau}_t; \xi_{t+1}) \mid \mathcal{G}_t] - D_t = \sum_{i,j} p_t(i) q_t(j)\, D_{t+1}(i, j; \xi_{t+1}) - D_t,$$

with $\tau_t \sim p_t$ and $\tilde{\tau}_t \sim q_t$ independent. In the closed probe,

$$\mathbb{E}\big[X_t^{\mathrm{closed}} \mid \mathcal{G}_t\big] = \mathbb{E}[D_{t+1}(\tau_t, \tau_t; \xi_{t+1}) \mid \mathcal{G}_t] - D_t = \sum_i p_t(i)\, D_{t+1}(i, i; \xi_{t+1}) - D_t.$$

Subtracting these two displays cancels the common $-D_t$ term (this is exactly why we needed to note $D_t$ is $\mathcal{F}_t$–measurable). Thus

$$\mathbb{E}[X_t^{\mathrm{open}} \mid \mathcal{G}_t] - \mathbb{E}[X_t^{\mathrm{closed}} \mid \mathcal{G}_t] = \sum_{i,j} p_t(i) q_t(j)\, D_{t+1}(i, j; \xi_{t+1}) - \sum_i p_t(i)\, D_{t+1}(i, i; \xi_{t+1}).$$

Finally, apply the tower property $\mathbb{E}[\cdot \mid \mathcal{F}_t] = \mathbb{E}(\mathbb{E}[\cdot \mid \mathcal{G}_t] \mid \mathcal{F}_t)$ to obtain

$$\mathbb{E}[X_t^{\mathrm{open}} \mid \mathcal{F}_t] - \mathbb{E}[X_t^{\mathrm{closed}} \mid \mathcal{F}_t] = \mathbb{E}[D_{t+1}(\tau_t, \tilde{\tau}_t; \xi_{t+1}) - D_{t+1}(\tau_t, \tau_t; \xi_{t+1}) \mid \mathcal{F}_t],$$

which by definition is $\Delta_t$. □

## A.4 CONTROLLED RANDOMIZATION NETWORK

**Lemma 5** (CRN antisymmetry and conditional mean). *Let the three–arm CRN evolve rollouts* $(+, -, 0)$ *with a common coupling of all non–token randomness. For a token pair* $(a, b)$ *let* $D_{t+1}^{\pm}(a, b)$ *denote the divergence at time* $t + 1$ *when the* $\pm$ *trajectory consumes* $(a, b)$ *at time* $t$. *Assume* $D_t^{\pm}$ *and* $D_{t+1}^{\pm}(a, b)$ *are integrable and* $\mathcal{F}_t$*–measurable as functions of* $(a, b)$. *Let* $\tau_t^{\pm} \sim p_t^{\pm}$ *and* $\tilde{\tau}_t^{\pm} \sim q_t^{\pm}$ *be conditionally independent given* $\mathcal{F}_t$. *By convention,*

$$D_{t+1}^{\pm} := D_{t+1}^{\pm}(\tau_t^{\pm}, \tilde{\tau}_t^{\pm}).$$

*Define*

$$D_{t+1,\text{closed}}^{\pm} := D_{t+1}^{\pm}(\tau_t^{\pm}, \tau_t^{\pm}), \qquad \Delta_t^{\pm} := \mathbb{E}\big[D_{t+1}^{\pm}(\tau_t^{\pm}, \tilde{\tau}_t^{\pm}) - D_{t+1}^{\pm}(\tau_t^{\pm}, \tau_t^{\pm}) \,\big|\, \mathcal{F}_t\big],$$

*and the CRN increments*

$$X_t := \tfrac{1}{2}\Big[(D_{t+1}^{+} - D_t^{+}) - (D_{t+1}^{-} - D_t^{-})\Big], \qquad X_{t,\text{closed}}^{\pm} := D_{t+1,\text{closed}}^{\pm} - D_t^{\pm}.$$

*Then:*

    *(i)* Antisymmetry. *Swapping* $+ \leftrightarrow -$ *maps* $X_t$ *to* $-X_t$.

    *(ii)* Conditional mean.

$$\mathbb{E}[X_t \mid \mathcal{F}_t] = \tfrac{1}{2}\Big(\mathbb{E}[X_{t,\text{closed}}^{+} \mid \mathcal{F}_t] - \mathbb{E}[X_{t,\text{closed}}^{-} \mid \mathcal{F}_t]\Big) + \tfrac{1}{2}\big(\Delta_t^{+} - \Delta_t^{-}\big).$$

    *(iii)* Neutrality and symmetry. *If closed–probe neutrality holds, meaning*

$$\mathbb{E}[X_{t,\text{closed}}^{\pm} \mid \mathcal{F}_t] = 0,$$

*then the first bracket in (ii) vanishes and one obtains*

$$\mathbb{E}[X_t \mid \mathcal{F}_t] = \tfrac{1}{2}(\Delta_t^{+} - \Delta_t^{-}).$$

*If in addition the open kernel is sign–symmetric, so that* $\Delta_t^{+} = \Delta_t^{-}$, *then the right–hand side is zero and hence*

$$\mathbb{E}[X_t \mid \mathcal{F}_t] = 0.$$

*Proof.* (i) is immediate: swapping $+ \leftrightarrow -$ exchanges the two terms in $X_t$, hence $X_t \mapsto -X_t$.

For (ii), $D_t^{\pm}$ are $\mathcal{F}_t$–measurable, so

$$\mathbb{E}[X_t \mid \mathcal{F}_t] = \tfrac{1}{2}\Big(\mathbb{E}[D_{t+1}^{+} \mid \mathcal{F}_t] - \mathbb{E}[D_{t+1}^{-} \mid \mathcal{F}_t]\Big) - \tfrac{1}{2}(D_t^{+} - D_t^{-}).$$

By Lemma 4 applied separately to $\{+, -\}$, we have

$$\mathbb{E}[D_{t+1}^{\pm} \mid \mathcal{F}_t] = \mathbb{E}[D_{t+1,\text{closed}}^{\pm} \mid \mathcal{F}_t] + \Delta_t^{\pm}.$$

Substituting gives

$$\mathbb{E}[X_t \mid \mathcal{F}_t] = \tfrac{1}{2}\Big(\mathbb{E}[D_{t+1,\text{closed}}^{+} \mid \mathcal{F}_t] - \mathbb{E}[D_{t+1,\text{closed}}^{-} \mid \mathcal{F}_t]\Big) + \tfrac{1}{2}(\Delta_t^{+} - \Delta_t^{-}) - \tfrac{1}{2}(D_t^{+} - D_t^{-})$$

$$= \tfrac{1}{2}\Big(\mathbb{E}[D_{t+1,\text{closed}}^{+} - D_t^{+} \mid \mathcal{F}_t] - \mathbb{E}[D_{t+1,\text{closed}}^{-} - D_t^{-} \mid \mathcal{F}_t]\Big) + \tfrac{1}{2}(\Delta_t^{+} - \Delta_t^{-}),$$

since $D_t^{\pm}$ are $\mathcal{F}_t$–measurable. Recognizing $X_{t,\text{closed}}^{\pm} = D_{t+1,\text{closed}}^{\pm} - D_t^{\pm}$, we obtain

$$\mathbb{E}[X_t \mid \mathcal{F}_t] = \tfrac{1}{2}\Big(\mathbb{E}[X_{t,\text{closed}}^{+} \mid \mathcal{F}_t] - \mathbb{E}[X_{t,\text{closed}}^{-} \mid \mathcal{F}_t]\Big) + \tfrac{1}{2}(\Delta_t^{+} - \Delta_t^{-}),$$

which is the claimed identity.

For (iii), under closed–probe neutrality both expectations $\mathbb{E}[X_{t,\text{closed}}^{\pm} \mid \mathcal{F}_t]$ vanish, so only the difference of open–kernel terms remains:

$$\mathbb{E}[X_t \mid \mathcal{F}_t] = \tfrac{1}{2}(\Delta_t^{+} - \Delta_t^{-}).$$

If moreover the open kernel is sign–symmetric, then $\Delta_t^{+} = \Delta_t^{-}$ and the conditional mean vanishes. $\square$

**Lemma 6** (Sibling averaging). *Let $\{X_t^{(m)}\}_{m=1}^M$ be conditionally i.i.d. CRN increments given $\mathcal{F}_t$ with $\mathbb{E}[|X_t^{(1)}| \mid \mathcal{F}_t] < \infty$. Then*

$$\overline{X}_t = \tfrac{1}{M} \sum_{m=1}^M X_t^{(m)} \xrightarrow{a.s.} \mathbb{E}[X_t^{(1)} \mid \mathcal{F}_t] \quad \text{as } M \to \infty.$$

*Proof.* Condition on $\mathcal{F}_t$. Given $\mathcal{F}_t$, the $X_t^{(m)}$ are i.i.d. with finite mean. By the strong law of large numbers (see (Kallenberg, 1997) for a more detailed proof), $\overline{X}_t \to \mathbb{E}[X_t^{(1)} \mid \mathcal{F}_t]$ almost surely for the conditional law, hence almost surely under $\mathbb{P}$. $\square$

### A.5 FILTRATION AND MEAN-FIELD PRELIMINARIES

**Definition 1** (Filtration). *$\mathcal{F}_t$ is the $\sigma$-algebra generated by hidden states, token draws, and CRN couplings up to step $t$.*

**Lemma 7** (Exchangeability). *If $\{X_t^{(i)}\}_{i \geq 1}$ is exchangeable with $\mathbb{E}[X_t^{(i)} \mid \mathcal{F}_t] = 0$ and $\mathbb{E}[|X_t^{(i)}|] < \infty$, then*

$$\frac{1}{N} \sum_{i=1}^N X_t^{(i)} \xrightarrow{a.s.} 0 \quad \text{as } N \to \infty.$$

*Proof.* By de Finetti's representation, exchangeable sequences are mixtures of i.i.d.; apply the SLLN inside the mixture and integrate (Kallenberg, 1997, Sec. 14). $\square$

## B PREDICTABLE DRIFT CORRIDOR

**Lemma 8** (Mean value theorem (JS)). *Fix $t$ and $i, j \in [V]$. Define $g_{t,i}(r) := \mathrm{JS}(\Phi_t(i), r)$ for $r \in \Delta^{V-1}$. Then for some $\theta \in [0, 1]$,*

$$\mathrm{JS}(\Phi_t(i), \widetilde{\Phi}_t(j)) - \mathrm{JS}(\Phi_t(i), \widetilde{\Phi}_t(i)) = \langle \nabla g_{t,i}(r_\theta), \, \widetilde{\Phi}_t(j) - \widetilde{\Phi}_t(i) \rangle,$$

*with $r_\theta = (1 - \theta)\widetilde{\Phi}_t(i) + \theta\widetilde{\Phi}_t(j)$. Hence*

$$\left| \mathrm{JS}(\Phi_t(i), \widetilde{\Phi}_t(j)) - \mathrm{JS}(\Phi_t(i), \widetilde{\Phi}_t(i)) \right| \leq L_{\mathrm{JS},t} \, \|\widetilde{\Phi}_t(j) - \widetilde{\Phi}_t(i)\|_2,$$

*where*

$$L_{\mathrm{JS},t} := \sup_{\substack{i,j \in [V] \\ \theta \in [0,1]}} \|\nabla_2 \mathrm{JS}(\Phi_t(i), r_\theta)\|_2.$$

**Lemma 9** (Lipschitz decoder and kernel). *For any $i, j \in [V]$,*

$$\|\widetilde{\Phi}_t(j) - \widetilde{\Phi}_t(i)\|_2 \; \leq \; L_{\mathrm{sm},t} \, \|W\|_2 \, L_{\mathrm{ker},t} \, \|E_j - E_i\|_2.$$

*If $\sigma_{\min}(W) > 0$, then*

$$\|\widetilde{\Phi}_t(j) - \widetilde{\Phi}_t(i)\|_2 \; \leq \; L_{\mathrm{sm},t} \, \kappa_2(W) \, L_{\mathrm{ker},t} \, \|M_j - M_i\|_2, \quad M = WE, \;\; \kappa_2(W) = \frac{\|W\|_2}{\sigma_{\min}(W)}.$$

**Proposition 3** (Predictable drift corridor). *Let $\mu_t = \mathbb{E}[X_t^{\mathrm{open}} \mid \mathcal{F}_t]$. Then*

$$|\mu_t| \; \leq \; L_{\mathrm{JS},t} \, L_{\mathrm{sm},t} \, L_{\mathrm{ker},t} \, \mathbb{E}_{i,j} \|E_j - E_i\|_2 \; =: \; c_t. \tag{7}$$

*If $\sigma_{\min}(W) > 0$, then*

$$|\mu_t| \; \leq \; L_{\mathrm{JS},t} \, L_{\mathrm{sm},t} \, \kappa_2(W) \, L_{\mathrm{ker},t} \, \mathbb{E}_{i,j} \|M_j - M_i\|_2. \tag{8}$$

*Proof.* Combine Lemma 8 with Lemma 9, then take expectation over $i \sim p_t$, $j \sim q_t$. This yields equation 4. The strengthened form equation 5 follows from $\|Wv\|_2 \geq \sigma_{\min}(W)\|v\|_2$. $\square$

## C    BLENDED REPORTING RULE

We collect here a complete derivation of the blended neutrality reporting bound used in the main text.

**Definition 2** (Centered increments, quadratic variation). *Let $X_t^{\text{open}} := D_{t+1} - D_t$, $\mu_t := \mathbb{E}[X_t^{\text{open}} \mid \mathcal{F}_t]$, and $Y_t := X_t^{\text{open}} - \mu_t$. Define*

$$M_N := \sum_{t=1}^{N} Y_t, \qquad B_N := \sum_{t=1}^{N} \mu_t, \qquad V_N := \sum_{t=1}^{N} \mathbb{E}[Y_t^2 \mid \mathcal{F}_{t-1}],$$

*and $\bar{X}_N := \frac{1}{N} \sum_{t=1}^{N} X_t^{\text{open}}$.*

### C.1    DETERMINISTIC EXPECTATION CONTROL

**Lemma 10** (Deterministic expectation control). *With $c_t$ as in equation 4,*

$$\left| \mathbb{E}[\bar{X}_N] \right| \leq \frac{1}{N} \sum_{t=1}^{N} c_t.$$

*Proof.* We have $S_N := \sum_{t=1}^{N} X_t^{\text{open}} = M_N + B_N$ by definition, so $\mathbb{E}[S_N] = \mathbb{E}[B_N]$. Therefore

$$\left| \mathbb{E}[\bar{X}_N] \right| = \frac{1}{N} \left| \mathbb{E}[B_N] \right| \leq \frac{1}{N} \sum_{t=1}^{N} \mathbb{E}[\,|\mu_t|\,] \leq \frac{1}{N} \sum_{t=1}^{N} c_t,$$

using equation 4. $\qquad\square$

### C.2    FREEDMAN PREREQUISITES AND DEVIATION

**Lemma 11** (Freedman prerequisites). *Under Lemma 3 and equation 4 there exists $c < \infty$ with $|Y_t| \leq c$ a.s., and $M_N$ is a martingale with predictable quadratic variation $V_N$.*

*Proof.* By Lemma 3, $|X_t^{\text{open}}| \leq \log 2$ a.s. and by equation 4, $|\mu_t| \leq c_t$. Let $c := \log 2 + \sup_s c_s < \infty$. Then $|Y_t| \leq |X_t^{\text{open}}| + |\mu_t| \leq c$. Measurability and $\mathbb{E}[Y_t \mid \mathcal{F}_{t-1}] = 0$ are by definition of $\mu_t$, so $\{M_t, \mathcal{F}_t\}$ is a martingale and $V_N$ is its predictable quadratic variation. $\qquad\square$

**Theorem 3** (Two-sided high-probability deviation). *For any $\delta \in (0, 1)$,*

$$|M_N| \;\leq\; \sqrt{2V_N \log(2/\delta)} + \tfrac{c}{3} \log(2/\delta) \quad \text{with probability at least } 1 - \delta,$$

*where $c$ is from Lemma 11. Equivalently,*

$$\left| \bar{X}_N - \frac{B_N}{N} \right| \leq \sqrt{\frac{2V_N \log(2/\delta)}{N^2}} + \frac{c}{3} \frac{\log(2/\delta)}{N}. \tag{9}$$

*Proof.* Apply Freedman's inequality to the martingale $M_N$ with bounded increments $|Y_t| \leq c$ (Lemma 11). Divide by $N$. $\qquad\square$

**Lemma 12** (Lindeberg condition). *Assume $V_N \to \infty$ in probability. Then for every $\epsilon > 0$,*

$$\frac{1}{V_N} \sum_{t=1}^{N} \mathbb{E}\Big[ Y_t^2 \, \mathbf{1}\{|Y_t| > \epsilon \sqrt{V_N}\} \,\Big|\, \mathcal{F}_{t-1} \Big] \xrightarrow{\mathbb{P}} 0.$$

*Proof.* Since $|Y_t| \leq c$, on $\{\sqrt{V_N} \geq c/\epsilon\}$ each indicator vanishes. As $V_N \to \infty$ in probability, the event holds with probability tending to one, so the normalized sum converges to 0 in probability. $\quad\square$

**Theorem 4** (Martingale Central Limit Theorem). *If $V_N/N \to \sigma^2 \in (0, \infty)$ in probability, then*

$$\frac{M_N}{\sqrt{V_N}} \Rightarrow \mathcal{N}(0, 1), \qquad \sqrt{N}\left( \bar{X}_N - \frac{B_N}{N} \right) \Rightarrow \mathcal{N}(0, \sigma^2).$$

*Proof.* By Lemma 12, Lindeberg's condition holds. The martingale central limit theorem yields $M_N/\sqrt{V_N} \Rightarrow \mathcal{N}(0,1)$; Slutsky gives the second convergence. $\square$

**Theorem 5** (Blended neutrality). *With $c_t$ from equation 4,*

$$\left|\mathbb{E}[\bar{X}_N]\right| \;\leq\; \min\left\{\frac{1}{N}\sum_{t=1}^{N}c_t,\; \left|\bar{X}_N - \frac{1}{N}\sum_{t=1}^{N}\mu_t\right| + z_{0.975}\frac{\widehat{s}_N}{\sqrt{N}}\right\},$$

*where $\widehat{s}_N^2 = \frac{1}{N}\sum_{t=1}^{N}(X_t^{\mathrm{open}} - \bar{X}_N)^2$. If $\frac{1}{N}\sum_{t=1}^{N}c_t \to 0$, then $\frac{1}{N}\sum_{t=1}^{N}\mu_t \to 0$ and the standard error band applies directly to $\bar{X}_N$.*

*Proof.* The first term inside the minimum is Lemma 10. For the second term, apply Theorem 3 to bound $|\bar{X}_N - B_N/N|$ in finite samples, or Theorem 4 to obtain the asymptotic normal band; replace the (unknown) variance by $\widehat{s}_N^2$ under the usual consistency. If $\frac{1}{N}\sum c_t \to 0$, then $B_N/N \to 0$, hence the band centers on $\bar{X}_N$ itself. $\square$

# D  MARKOV KERNEL DRIFT AND CORRIDOR BOUNDS

This appendix collects the kernel-level derivations underlying Proposition 1 in Section 3.4. We assume that each residual block $H_\ell(x) = x + G_\ell(\mathrm{LN}(x))$ is Lipschitz with constant $L_\ell$, so that the cumulative kernel constant satisfies $L_{\mathrm{ker},t} = \prod_{\ell \leq t} L_\ell$. This assumption is standard in theoretical analyses of residual networks (Hardt & Ma, 2017; Hayou et al., 2019; Tian, 2017) and is used only as a structural input to the corridor bound.

**Definition 3** (Open probe kernel). *At step $t$, condition on $\mathcal{F}_t$, which fixes the paired hidden states $(h_t, \tilde{h}_t)$ and decoded distributions $(p_t, q_t)$. Let $\xi_{t+1}$ denote the exogenous randomness used by the one–step transition. The open-probe kernel acts on a token pair $(i,j) \in [V]^2$ as*

$$D_{t+1}(i,j;\xi_{t+1}) := \mathrm{JS}\big(S(WK(h_t,i;\xi_{t+1})), S(WK(\tilde{h}_t,j;\xi_{t+1}))\big),$$

*with $S$ the softmax, $W$ the decoder, and $K$ the kernel map.*

**Lemma 13** (Drift identity). *For the open probe increment $X_t^{\mathrm{open}} = D_{t+1} - D_t$, the predictable mean satisfies*

$$\mu_t = \mathbb{E}_{i\sim p_t,\; j\sim q_t}\big[D_{t+1}(i,j;\xi_{t+1}) - D_{t+1}(i,i;\xi_{t+1})\,\big|\,\mathcal{F}_t\big]. \tag{10}$$

*Proof.* Condition on $\mathcal{F}_t$ and expand the definition of $X_t^{\mathrm{open}}$. The baseline term corresponds to both arms sampling $i \sim p_t$; the open probe uses independent $i \sim p_t$, $j \sim q_t$. Subtracting and taking conditional expectation yields equation 10. $\square$

**Theorem 6** (Expected drift bound). *With notation as in Lemma 13,*

$$|\mu_t| \;\leq\; L_{\mathrm{JS},t}\, L_{\mathrm{sm},t}\, \|W\|_2\, L_{\mathrm{ker},t}\, \mathbb{E}_{i,j}\|E_j - E_i\|_2.$$

*If $\sigma_{\min}(W) > 0$, the strengthened logit–space version*

$$|\mu_t| \;\leq\; L_{\mathrm{JS},t}\, L_{\mathrm{sm},t}\, \kappa_2(W)\, L_{\mathrm{ker},t}\, \mathbb{E}_{i,j}\|M_j - M_i\|_2, \qquad \kappa_2(W) = \frac{\|W\|_2}{\sigma_{\min}(W)},$$

*also holds.*

*Proof.* Fix $i,j$ and apply the mean value theorem to $r \mapsto \mathrm{JS}(\Phi_t(i), r)$ with $\Phi_t(i) = S(WK(h_t,i))$, $\widetilde{\Phi}_t(j) = S(WK(\tilde{h}_t,j))$. This yields

$$|\mathrm{JS}(\Phi_t(i), \widetilde{\Phi}_t(j)) - \mathrm{JS}(\Phi_t(i), \widetilde{\Phi}_t(i))| \leq L_{\mathrm{JS},t}\,\|\widetilde{\Phi}_t(j) - \widetilde{\Phi}_t(i)\|_2.$$

Bound the difference $\widetilde{\Phi}_t(j) - \widetilde{\Phi}_t(i)$ by the composition of Lipschitz constants for softmax, decoder, and kernel (Appendix A.1–A.1). Taking expectation over $i \sim p_t$, $j \sim q_t$ gives the bound. If $\sigma_{\min}(W) > 0$, replace $\|E_j - E_i\|_2$ by $\|M_j - M_i\|_2$ to obtain the strengthened form. $\square$

## E  MEAN-FIELD LIFT OF NEUTRALITY

This appendix provides the rigorous proof of Theorem 2, showing that neutrality and the blended reporting rule persist in the mean-field limit.

**Definition 4** (Exchangeability). *A collection of random variables $\{A_k\}_{k=1}^N$ is exchangeable if its joint distribution is invariant under finite permutations. In our setting, the "agents" $A_k$ are either:*

1. *token pairs $(i, j)$ drawn from $(p_t, q_t)$ at a fixed step $t$ (*trajectory view*), or*

2. *residual blocks $H_\ell$ contributing finite-difference drifts (*layerwise view*).*

**Lemma 14** (Law of large numbers for exchangeable agents). *Let $\{A_k\}_{k=1}^N$ be exchangeable with $\mathbb{E}[A_1] = 0$ and $\mathrm{Var}(A_1) < \infty$. Then*

$$\frac{1}{N} \sum_{k=1}^N A_k \xrightarrow{P} 0 \qquad as\ N \to \infty.$$

*Proof.* By de Finetti's representation, exchangeable sequences are mixtures of i.i.d. sequences. Apply the strong law of large numbers conditionally, then integrate over the mixing measure to obtain convergence in probability. □

**Theorem 7** (Mean-field neutrality). *Fix a time $t$. Let $\{X_{t,a}^{\mathrm{open}}\}_{a=1}^M$ be the agent actions (either in the trajectory or layerwise view), assumed exchangeable and integrable, with $|X_{t,a}^{\mathrm{open}}| \le b$ almost surely (cf.Lemma 3). If agent-level neutrality holds, i.e.*

$$\mathbb{E}[X_{t,a}^{\mathrm{open}} \mid \mathcal{F}_t] = 0 \quad for\ all\ a,$$

*then*

$$\frac{1}{M} \sum_{a=1}^M X_{t,a}^{\mathrm{open}} \xrightarrow[M \to \infty]{a.s.} 0.$$

*Consequently, the population law inherits neutrality. Moreover, because $|X_{t,a}^{\mathrm{open}}| \le b$ and $|\mu_t| \le c_t$ (Theorem 6), the predictable-corridor and blended-reporting bounds (Theorem 1; Appendix C) hold unchanged in the mean-field limit.*

*Proof.* By exchangeability of $\{X_{t,a}^{\mathrm{open}}\}_{a \ge 1}$ there exists a directing random measure $\Lambda_t$ such that, conditional on $\mathcal{G}_t := \sigma(\mathcal{F}_t, \Lambda_t)$, the sequence is i.i.d. (de Finetti; cf. Lemma 7). Since $|X_{t,a}^{\mathrm{open}}| \le b$ and $\mathbb{E}[X_{t,a}^{\mathrm{open}} \mid \mathcal{F}_t] = 0$ by assumption, we also have $\mathbb{E}[X_{t,a}^{\mathrm{open}} \mid \mathcal{G}_t] = 0$. Applying the strong law of large numbers conditionally on $\mathcal{G}_t$ yields

$$\frac{1}{M} \sum_{a=1}^M X_{t,a}^{\mathrm{open}} \xrightarrow[M \to \infty]{a.s.} \mathbb{E}[X_{t,1}^{\mathrm{open}} \mid \mathcal{G}_t] = 0.$$

Thus, the empirical mean converges almost surely to zero, and the population law inherits neutrality. Finally, the corridor bound $|\mu_t| \le c_t$ depends only on architectural constants and embeddings, so it is unaffected by averaging. Uniform boundedness of the increments (Lemma equation 3) ensures that the Freedman/CLT arguments in Appendix C apply unchanged, so the blended reporting rule extends to the mean-field limit. □

## F  ABLATION STUDIES

To test the robustness of our neutrality results we vary two key hyperparameters: the sampling temperature $T$ and the number of siblings $M$. Lower and higher temperatures alter output entropy, while $M$ controls the variance reduction from sibling averaging. Across all settings, closed probes continue to behave as martingale differences, and open probes remain corridor-bounded. The reported $E_{\max}$ values in Tables 5 stay close to one, which indicates flat $e$-processes. Importantly, neutrality is only rejected if $E_{\max}$ exceeds $1/\alpha \approx 20$ at $\alpha = 0.05$, so values such as $E_{\max} = 1.353$ are well within the neutrality region and reflect no systematic drift.

| Setting | Model | Probe | Mean drift | $t$-test $p$ | Emax |
|---------|-------|-------|-----------|------------|------|
| T=0.5, M=16 | gpt2-medium | Closed | $-4.026e-9$ | $3.17e-01$ | 1.022 |
| T=0.5, M=16 | gpt2-medium | Open | $3.060e-9$ | $3.17e-01$ | 1.022 |
| T=1, M=16 | gpt2-medium | Closed | $-2.742e-2$ | $6.34e-01$ | 1.118 |
| T=1, M=16 | gpt2-medium | Open | $-8.534e-9$ | $6.34e-01$ | 1.118 |
| T=2, M=16 | gpt2-medium | Closed | $-9.122e-4$ | $7.70e-02$ | 1.186 |
| T=2, M=16 | gpt2-medium | Open | $1.510e-9$ | $7.70e-02$ | 1.186 |
| T=5, M=16 | gpt2-medium | Closed | $6.957e-4$ | $9.90e-02$ | 1.004 |
| T=5, M=16 | gpt2-medium | Open | $3.861e-8$ | $9.90e-02$ | 1.004 |
| T=1.7, M=8 | gpt2-medium | Closed | $-3.176e-4$ | $7.94e-01$ | 1.005 |
| T=1.7, M=8 | gpt2-medium | Open | $-1.127e-8$ | $7.94e-01$ | 1.005 |
| T=1.7, M=4 | gpt2-medium | Closed | $1.985e-3$ | $9.79e-01$ | 2.025 |
| T=1.7, M=4 | gpt2-medium | Open | $5.077e-9$ | $9.79e-01$ | 2.025 |

Table 5: Ablation neutrality audits for gpt2-medium under varying temperature $T$ and sibling count $M$. Closed probes show centered but wandering cumulative drift; open probes remain numerically tiny by comparison. $E_{\max}$ values near 1 indicate no evidence of sustained bias; for context, an anytime $e$-test would only approach rejection around $E_{\max} \gtrsim 20$ at $\alpha = 0.05$.

| Model | Block | $\hat{\mu}$ | SE | 95% CI |
|-------|-------|------|-----|--------|
| tiny-gpt2 | All (4) | $1.157e-07$ | $9.729e-08$ | $(-1.877e-10, 3.468e-07)$ |
| distilgpt2 | All (4) | $-2.833e-04$ | $9.159e-03$ | $(-1.839e-02, 1.843e-02)$ |
| gpt2-medium | All (4) | $-2.844e-04$ | $8.178e-03$ | $(-1.687e-02, 1.630e-02)$ |
| gpt2-large | All (4) | $1.221e-04$ | $1.341e-02$ | $(-2.851e-02, 2.788e-02)$ |

Table 6: Layer-as-agent diagnostics aggregated across residual blocks. Reported are the mean action $\hat{\mu}$, its standard error, and a 95% confidence interval. Intervals cover zero throughout, indicating no systematic bias at the block level.

## F.1 LAYER PERSPECTIVE.

## F.2 TRAJECTORY-LEVEL NEUTRALITY AUDITS WITH RESPECT TO $T$

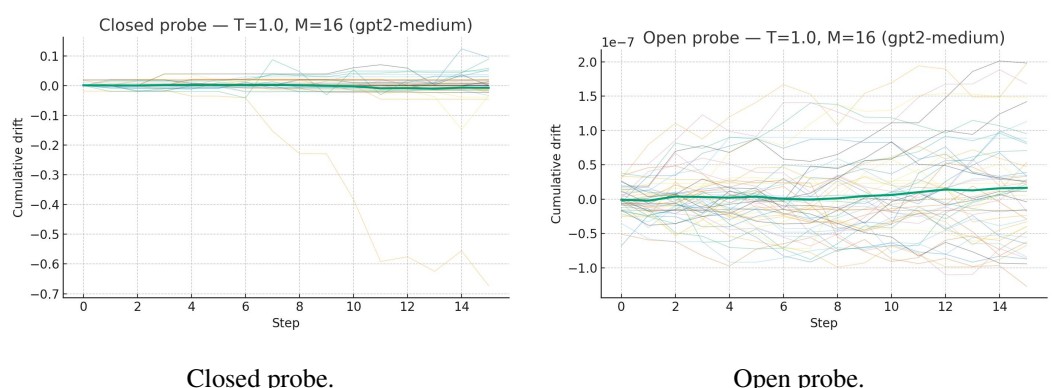

Closed probe.                              Open probe.

Figure 3: Neutrality audit for gpt2-medium with $T = 1, M = 16$.

**Remark regarding $T = 5$.** At high temperature the softmax flattens, increasing token entropy and branching variance in the open probe; closed increments remain martingale differences, but their step variance also grows because re-embeddings explore more of the state space. In Table 5 ($T=5$, $M=16$) the prompt-level $t$-test is marginal ($p = 3.20\times10^{-2}$) around a very small mean drift ($6.19\times10^{-5}$), yet the anytime $e$-test stays near one ($E_{\max} = 1.005$), far below rejection thresholds (e.g., $\geq 20$ at $\alpha=0.05$), indicating no sustained deviation. Trajectories therefore look more volatile (variance inflation) but remain neutral in expectation. Moreover, theory predicts a smaller corridor at higher $T$ (softmax Lipschitz $1/(2T)$), consistent with the absence of bias despite noisier paths.

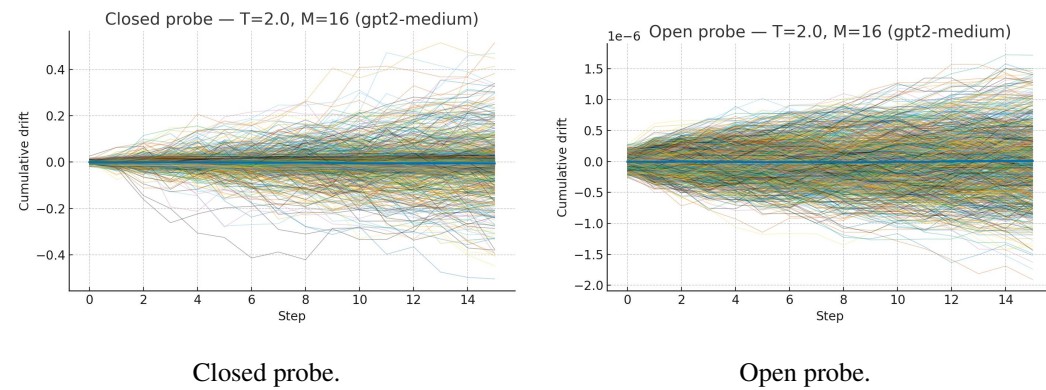

Closed probe.            Open probe.

Figure 4: Neutrality audit for `gpt2-medium` with $T = 2, M = 16$.

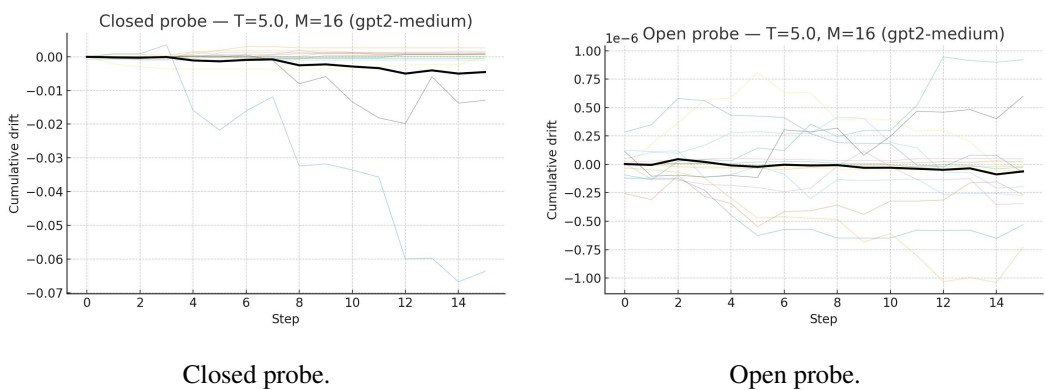

Closed probe.            Open probe.

Figure 5: Neutrality audit for `gpt2-medium` with $T = 5, M = 16$.

### F.3 TRAJECTORY-LEVEL NEUTRALITY AUDITS WITH RESPECT TO $M$

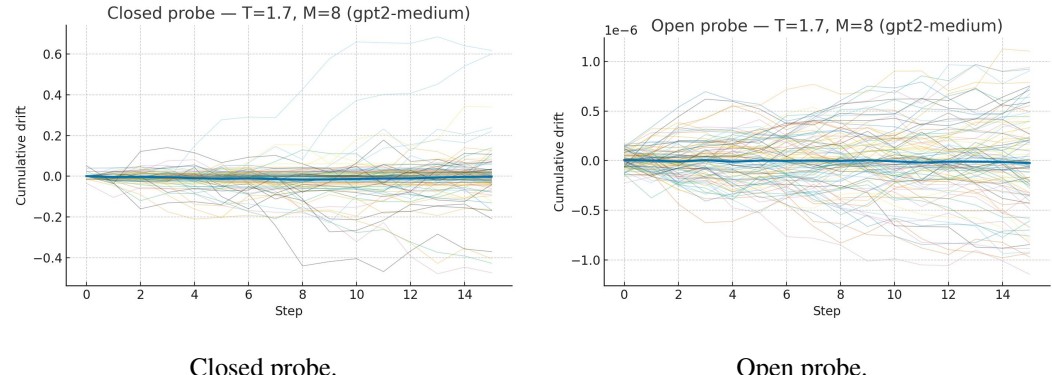

Closed probe.            Open probe.

Figure 6: Neutrality audit for `gpt2-medium` with $T = 1.7, M = 8$.

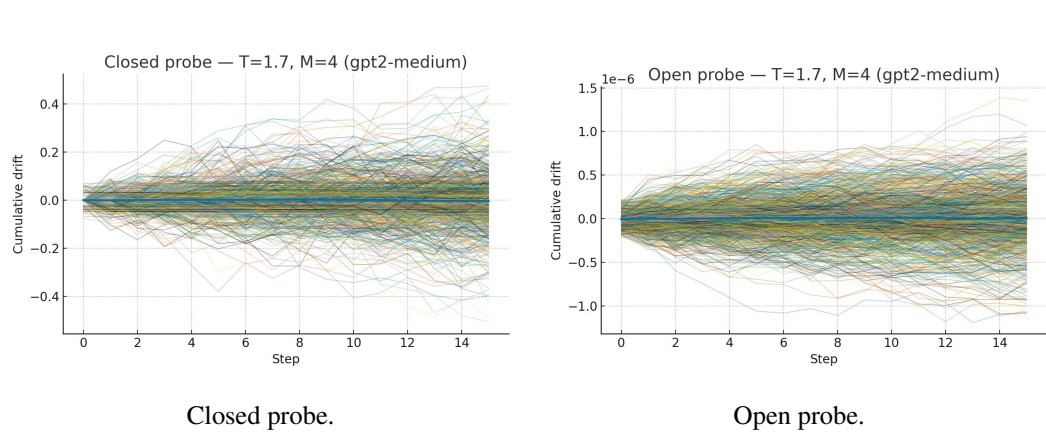

Closed probe.                              Open probe.

Figure 7: Neutrality audit for gpt2-medium with $T = 1.7, M = 4$.

