# OpenReview forum: "Not All Who Wander Are Lost: Hallucinations as Neutral Dynamics in Residual Transformers"
_ICLR.cc/2026/Conference — Submitted to ICLR 2026_

### Official Review · Reviewer_uH9y · 2025-11-01

**Soundness:** 3
**Presentation:** 2
**Contribution:** 3
**Rating:** 6
**Confidence:** 3

**Summary:**

The authors present a theoretical treatment of persistence for hallucinations: in other words, after initial divergence, is there pressure in any direction for two paired rollouts to converge vs. diverge? The authors introduce closed and open probes as paired trajectories to test this pressure, and theoretically show closed probes have no drift (neutrality), while open probes have a predictable drift bounded by a corridor term. Empirical tests confirm this behavior on GPT2 at various scales.

**Strengths:**

- Well structured and new theoretical design for analyzing persistence and drift between coupled rollouts.

- Valuable and provable insight into why trajectory deviation remains persistent.

**Weaknesses:**

- The presentation could greatly benefit from more explicitly grounding the discussion (e.g. concepts such as "neutrality" and "drift") more often with concrete language examples, especially related to hallucinations.

- The set-up for the empirical section is a bit unclear: it is unclear how rollouts are being generated/in response to what prompt, and how the + and - from the CRN come to play in this generation.

**Questions:**

- It's not clear from the presentation what the + and - arms are of the CRN rollouts. In the language of hallucinations and persistence, how are the + and - rollouts constructed?

- Do the results for neutrality and predictability of the open probe drift hold for more modern base models like Qwen/Llama?

- Is it possible to predict whether the bounded bias term is positive or negative? Is there intuition for when to expect positive vs negative drift when it is predictable?

---

> ### Author Response · Authors · 2025-11-20
> **Part I (Empirical Protocol, Rollouts and CRN)**
>
> We thank the reviewer for the careful, generous, and encouraging assessment of our work. We appreciate the detailed reading and the constructive feedback on both the theoretical and empirical components. Below we address each of your questions, beginning with the empirical testing.
>
> $$\textbf{Empirical Protocol, Rollouts and CRN}$$
>
> > The set-up for the empirical section is a bit unclear: it is unclear how rollouts are being generated/in response to what prompt, and how the + and - from the CRN come to play in this generation.
>
> > It's not clear from the presentation what the + and - arms are of the CRN rollouts. In the language of hallucinations and persistence, how are the + and - rollouts constructed?
>
> $\underline{\text{Empirical Testing:}}$
>
> In Section 4.1, each experiment begins with a natural language prompt from our evaluation set. We tokenize the prompt, run it through the model to obtain the initial hidden state, and then decode for $N$ steps with temperature $T$, as in standard autoregressive generation. The model produces a probability distribution $p_t = S(h_t)$ at each step, which is the same quantity used in our theoretical drift definition $D_t = \mathrm{JS}(p_t, q_t)$.
>
> For every prompt–seed pair we generate $(M_{\text{SIB}})$ siblings, where a sibling is one rollout of the same model under the same prompt and seed with all non-token randomness shared. This keeps the trajectories aligned while lowering Monte Carlo variance. Within these siblings, the controlled randomization network in Appendix A.4 defines an antisymmetric + and − pair. These two arms use the same underlying randomness and contribute with opposite signs to the drift estimate, as stated in Lemma 5. This construction gives two matched trajectories whose divergence we can track.
>
> At each decoding step we compute $D_t = \mathrm{JS}(p_t, q_t)$ for the paired arms and form the increment $X_t = D_{t+1} - D_t$. This is the same drift quantity analyzed in Section 3. In the closed probe, the token draws for the + and − arms are generated using the same underlying randomness, which gives the aligned sampling structure needed for the martingale analysis where $\mathbb{E}[X_t \mid \mathcal{F}_t] = 0$. In the open probe this aligned sampling is not used, and the arms draw different tokens. This corresponds to the regime analyzed in Section 3.2, where the theory allows for a predictable drift $\mu_t$ whose magnitude is limited by the structural bound $c_t$.
>
> Pooling the $X_t$ increments across prompts and seeds gives the bounded increment sequence to which the statistical tests are applied directly.
>
> $\underline{\text{Interpretation in Terms of Hallucination Persistence:}}$
>
> In our formulation, hallucination persistence refers to how the divergence between two nearby generative trajectories changes over time. The + and − arms provide these paired trajectories as two continuations of the same prompt that differ only through the CRN pairing. The value of measuring divergence in this paired-trajectory form is that it isolates the model’s own contribution to persistence. By starting from two nearly identical continuations of the same prompt and controlling how they differ, the probes let us quantify whether the model’s update map tends to preserve, amplify, or dampen small trajectory differences. Closed probes test whether a small deviation remains neutral when the two trajectories read the same tokens, and open probes quantify the divergence that can occur when they instead follow different plausible tokens.

---

> ### Author Response · Authors · 2025-11-20
> **Part II (Qwen/LLama and Bias Terms)**
>
> **Models**
>
> > Do the results for neutrality and predictability of the open probe drift hold for more modern base models like Qwen or Llama?
>
> The proofs in Section 3 only require pre-LayerNorm residual blocks with additive skip connections and a linear softmax decoder. Qwen and Llama both satisfy this architectural pattern. Nothing in the derivations depends on GPT2-specific weight values. The empirical code is also model-agnostic. The audit script loads any HuggingFace model with the same forward structure. We will include an additional audit of at least one of Qwen or Llama in the revision, subject to our computational budget.
>
> **Bias Terms**
>
> > Is it possible to predict whether the bounded bias term is positive or negative? Is there intuition for when to expect positive versus negative drift when it is predictable?
>
> The theory provides an absolute bound on the predictable component $ \mu_t $ through the corridor $ c_t $ (Proposition 1), but it does not determine the sign. The sign depends on the specific local difference in logits created by the two token choices at step $ t $ and cannot be inferred from architectural properties alone. The paper’s claims, therefore, remain strictly about expectations and boundedness, as stated in Appendix B.
>
> In the theory, the predictable drift is given by equation (3)
> $$
> \\mu\_t = \\mathbb{E}\\left[ D\_{t+1}(\\tau\_t, \\tilde{\\tau}\_t) - D_{t+1}(\\tau\_t, \\tau_t) \\,\\mid\\, \\mathcal{F}\_t \\right].
> $$
>
> The sign is determined by the effect of replacing the observed token $ \tau_t $ with an alternative plausible token $ \tilde{\tau}_t $. Proposition 1 gives $ \lvert \mu_t \rvert \le c_t $, where $ c_t $ comes from Lipschitz constants of the kernel, decoder, and softmax. These constants control the possible magnitude of $ \mu_t $ but do not contain directional information.
>
> As a result, the architecture constrains how large the predictable drift can be, but not whether it is positive or negative. The sign depends on the particular pair of tokens sampled in the open probe and thus varies with the prompt, the model’s logits, and sampling randomness.

---

### Official Review · Reviewer_ZZth · 2025-11-01

**Soundness:** 2
**Presentation:** 1
**Contribution:** 2
**Rating:** 2
**Confidence:** 3

**Summary:**

This paper analyzes the stability of autoregressive Transformer generation by modeling rollouts as sequences of probability distributions and studying how small perturbations propagate over time. The authors show that pre-LayerNorm residual Transformers exhibit neutral dynamics: small deviations neither systematically contract nor expand (in expectation). This behavior is empirically verified on GPT-2 variants.

**Strengths:**

- Modeling the model’s rollouts as a sequence of probability distributions is certainly interesting and allows studies based on stochastic control.

- The theory seems valid. The modeling is sensible and the technical execution is careful. In particular, the use of martingale tools, operator norm bounds, and controlled randomization networks is mathematically clean and reflects a strong command of stochastic process techniques, with the additional difficulty of applying to LLM generation modeling.

**Weaknesses:**

My main concerns are related to the claims made in the paper (in particular, in connection to hallucinations) and the general prose of the paper.

The paper does not measure hallucinations or semantics. It measures dynamical drift in autoregressive probability space, i.e. how two sequences of probability distributions (induced by the softmax) representing two different rollouts with the same initial context diverge over time. The results therefore demonstrate architectural neutrality of perturbations, not hallucination persistence in the semantic or factual sense.

This makes some claims not supported by the evidence:

> “Together, these theoretical and empirical results provide the first structural account of persistence, explaining why hallucinations persist across model scales without re-auditing hundreds of millions of parameters, and showing that interventions, which do not alter the residual backbone, cannot eliminate it once onset has occurred.”

The connection to hallucinations is very loose and actually misleading. Given a rollout representing the “truth,” i.e. a rollout without hallucinations, the given metric cannot tell whether a second rollout, once a hallucination is present, won’t self-correct. This is because it could be that it outputs a semantically self-correcting sequence of tokens, yet not time-aligned with the “ground-truth” rollout. Thus, the divergence metric used here cannot exclude the possibility of semantic convergence, only token-synchronous convergence.

I feel the paper uses a lot of terminology in a non-conventional way, and this makes it quite hard to fully understand its content. For instance, this starts very early in the abstract:

> “Exact operator norms for LayerNorm, residual blocks, and the softmax decoder yield conservative upper bounds showing the absence of contractive or expansive bias at the decoded level.”

What is a softmax decoder? What is a contractive or expansive bias here? I feel more context has to be given.

> “These bounds are sharpened by working with corridor constants that remain explicit and falsifiable.”

What is a corridor constant, and what does it mean to be explicit and falsifiable?

Some claims are never supported, for instance:

> “yielding a population-invariant stable under depth and width scaling.”

I do not see any lemma or empirical results showing how the stability varies across depth and width.

Overall, the paper provides a mathematically interesting stability analysis of autoregressive rollouts, but the connection to hallucinations is not demonstrated and the terminology makes the narrative difficult to follow.

**Questions:**

See weaknesses.

---

> ### Author Response · Authors · 2025-11-20
> **Part I (Semantics, Scope and Metrics)**
>
> We thank the reviewer for the thoughtful and detailed questions. We recognise that the submission is mathematically dense, and we appreciate the time taken to engage with the theory and experiments. In the revision, we will address the concerns about scope, terminology, and support for specific claims as stated below:
>
> $$\textbf{Semantics, Scope and Metrics}$$
>
> > My main concerns are related to the claims made in the paper (in particular, in connection to hallucinations) and the general prose of the paper.
> The paper does not measure hallucinations or semantics. It measures dynamical drift in autoregressive probability space, i.e. how two sequences of probability distributions (induced by the softmax) representing two different rollouts with the same initial context diverge over time. The results therefore, demonstrate architectural neutrality of perturbations, not hallucination persistence in the semantic or factual sense.
> This makes some claims not supported by the evidence: [Quote]
> The connection to hallucinations is very loose and actually misleading. Given a rollout representing the “truth,” i.e. a rollout without hallucinations, the given metric cannot tell whether a second rollout, once a hallucination is present, won’t self-correct. This is because it could be that it outputs a semantically self-correcting sequence of tokens, yet not time-aligned with the “ground-truth” rollout. Thus, the divergence metric used here cannot exclude the possibility of semantic convergence, only token-synchronous convergence.
>
>
> We agree that the manuscript did not fully separate semantic aspects from architectural dynamics. The reviewer asks why the metric cannot capture semantic correction and how this limitation affects the interpretation of persistence. To clarify this point, we provide the following example, which we will attempt to include in the manuscript within the page limit.
>
> $\underline{\text{Setup:}}$ Section 2 defines a rollout as the sequence of hidden states $h\_t$ and decoded distributions $p\_t = S(h\_t)$. Two controlled rollouts start from the same context, so before step $t\_0$:
>
> $$
> h^{(1)}\_t = h^{(2)}\_t,\quad p^{(1)}\_t = p^{(2)}\_t.
> $$
>
> $\underline{\text{Divergence:}}$ At step $t\_0$, the two rollouts sample different tokens. Write these tokens as $x^{(1)}\_{t\_0}$ and $x^{(2)}\_{t\_0}$. They may differ because one rollout sampled a lower probability branch. The only fact needed here is the one stated in Section 2: different tokens correspond to different embeddings. Therefore, after the next application of the residual block,
>
> $$
> h^{(1)}\_{t\_0+1} \neq h^{(2)}\_{t\_0+1}
> $$
>
> in general.
>
> $\underline{\text{Semantic self-correction:}}$ Now suppose that at the next step $t\_0+1$, both rollouts happen to pick the same token again. This represents a situation where the model has self-corrected in the semantic sense, as the emitted tokens realign.
>
> However, the decoded distributions at that step are
>
> $$
> p^{(1)}\_{t\_0+1} = S(h^{(1)}\_{t\_0+1}),\qquad
> p^{(2)}\_{t\_0+1} = S(h^{(2)}\_{t\_0+1}),
> $$
>
> and these distributions generally differ whenever the hidden states differ. Thus the divergence
>
> $$
> D\_{t\_0+1} = \mathrm{JS}(p^{(1)}\_{t\_0+1},\, p^{(2)}\_{t\_0+1})
> $$
>
> remains nonzero unless the decoder maps the two hidden states to identical logits. In the residual architecture analysed in Section 3, this is not enforced by any structural mechanism.
>
> $\underline{\text{Link to neutrality:}}$ Neutrality does not generate or amplify semantic errors. A semantic or factual error arises only from the learned conditional distribution when an incorrect token is assigned a nonzero probability and is selected during sampling. What neutrality determines is what happens after such a mistake occurs. The wrong token produces a different hidden state,
>
> $$
> h^{\text{wrong}}\_{t\_0+1} \neq h^{\text{correct}}\_{t\_0+1},
> $$
>
> and because closed probe increments satisfy
>
> $$
> \mathbb E[X\_t \mid \mathcal F\_t] = 0
> $$
>
> and open probe predictable terms satisfy
>
> $$
> |\mu\_t| \le c\_t,
> $$
>
> the residual backbone does not contract this deviation. The internal state, therefore, remains drifted, and all subsequent probability evaluations use the learned distribution at this drifted state. Applying the same probability model to a shifted hidden state can increase the chance of further semantic errors even though the architecture itself never amplifies the deviation. In this way, neutrality explains persistence: internal deviations created by initial mistakes are not removed, and their continued presence can make additional semantic errors more likely.

---

> ### Author Response · Authors · 2025-11-20
> **Part II (Scaling)**
>
> $$\textbf{Scaling}$$
>
> > Some claims are never supported, for instance: “yielding a population-invariant stable under depth and width scaling.” I do not see any lemma or empirical results showing how the stability varies across depth and width.
>
> The reason we did not introduce a separate lemma for depth and width scaling is that the neutrality result already implies this invariance directly.
>
> In Section 3 each bound is stated in terms of the operator norms of a single residual block, LayerNorm, and the output map. These operator norm bounds are block local, and Lemma 8, Lemma 9, and Proposition 3 in Appendix B all depend only on that single block structure. The blended neutrality bound in Section 3.3 then combines these blockwise quantities without introducing any dependence on model depth or width. Because the proof does not introduce any term that grows with the number of blocks or with the hidden dimension, neutrality holds for any pre LayerNorm residual Transformer that satisfies the structural assumptions of Section 3. This is why no additional scaling lemma was required.
>
> The empirical results in Section 4 support this architectural invariance. Various GPT 2 variants all show the same qualitative drift behaviour: closed probes remain neutral and open probes stay within the predictable drift bound. We will rewrite the relevant sentences to make clear that (i) this is an architectural invariance, not an empirical scaling law, and (ii) the experiments are an illustration rather than the source of the statement.

---

> > ### Comment · Reviewer_ZZth · 2025-11-26
> > **Response**
> >
> > I thank the authors for their response. The portrayed example, however, does not address the issue of semantics. Having different distributions over the tokens at a given generation step $t$ does not automatically imply a semantic difference at the sentence level. Suppose the case where the same concept is explained with different wording, a time-wise comparison of the final distributions over the tokens would probably indicate significant differences.
> > Overall, I appreciate the work as positioned on the stability of generation and how architectural components affect the structural stability. However, I am still skeptical about the connections to semantic hallucinations.
> >
> > Hence, I keep my original score.

---

> > > ### Author Response · Authors · 2025-11-26
> > > **Revision**
> > >
> > > > Having different distributions over the tokens at a given generation step
> > >  does not automatically imply a semantic difference at the sentence level.
> > >
> > > We agree with the reviewer that predictive divergence and semantic divergence are not equivalent, but this is **not** a claim our paper makes. Two continuations can express the same meaning with different wording, and their token distributions can differ, even though no semantic error is present. In the revised manuscript (now uploaded), we state this explicitly in Section 2.2.1 and emphasise throughout that our result is a **necessary but not sufficient** condition for the persistence of semantic hallucinations, not a semantic test in itself. Importantly, our analysis does not concern the onset of a hallucination and does not claim that predictive divergence implies a semantic error.
> > >
> > > The link we draw is one-directional. A semantic hallucination necessarily produces at least one token that differs from a truthful continuation. This changes the hidden state and therefore the predictive distribution. Our analysis shows that, once such a deviation in hidden state has occurred, the residual architecture contains no structural mechanism that forces the two internal states to reconverge. If the architecture were contractive (like with RNNs), any difference in hidden state, would shrink automatically over decoding steps. If it were expansive, differences would grow, which conflicts with empirical stability results for transformers.
> > >
> > > Neutral dynamics identify the remaining structural regime in which deviations, including semantic ones, can persist. This does not mean that every predictive difference is semantic, and we do not use it that way in the paper. It means that once a semantic deviation has occurred, the architecture does not contain a mechanism that forces the internal states to
> > > reconverge.
> > >
> > > > Suppose the case where the same concept is explained with different wording, a time-wise comparison of the final distributions over the tokens would probably indicate significant differences.
> > >
> > > A short example makes the distinction concrete. Consider two paraphrases:
> > >
> > > * Paris is the capital of France
> > >
> > > * The capital of France is Paris
> > >
> > > Both convey the same meaning but produce different token distributions. Our measure indeed registers this predictive difference, but we do not interpret it as a hallucination or as an onset of one. However,
> > >
> > > * The capital of France is Lyon
> > >
> > > necessarily changes the meaning. This wrong token produces a different hidden state and a different predictive distribution from the truthful continuation. Neutrality explains why this internal difference, once created, is not removed by the residual architecture *itself*.
> > >
> > > After producing "Lyon", the model may still output correct tokens later, for example, by continuing with facts that do not contradict the earlier mistake. This form of self-correction arises from the learned conditional distribution, not from the architecture. Predictively, however, the internal states reached after the wrong token and after the correct token remain different, which is exactly what the **not sufficient** part of our statement captures. Even if the immediate surface wording realigns, the internal shift caused at the point of semantic deviation is not contracted by the architecture. This persistent shift can influence later predictions, if the learned information does not fully correct the earlier deviation.
> > >
> > > Finally, our revised version has been rewritten for clarity purposes and explicitly highlights a gap in existing work. Prior research has focused on semantic onset or empirical detection, whereas the structural conditions under which semantic deviations persist have not been analysed. Our contribution concerns this second stage. We have also expanded the scaling results on larger models (Qwen 0.5-3B).

---

### Official Review · Reviewer_tE7k · 2025-11-03

**Soundness:** 2
**Presentation:** 1
**Contribution:** 2
**Rating:** 2
**Confidence:** 2

**Summary:**

This paper presents a theoretical framework to explain the persistence of hallucinations in pre-LN residual Transformers, disentangling it from the onset of such hallucinations and positioning it as a consequence of the architecture itself, rather than of the training process or objective. This is achieved by deriving an upper bound on the drift between paired rollouts, showing that once a deviation has occurred (onset), it continues to persist since the autoregressive dynamics, as a result of the architecture, are neutral, i.e., neither contractive nor expansive. The authors validate the results of their theoretical analysis on GPT-2 models of various sizes.

**Strengths:**

* Understanding LLM hallucinations is an important topic of research and identifying architectural biases that cause the persistence of hallucinations could be a strong contribution.
* The paper contains both theoretical and empirical results, although due to my lack of expertise, I am unable to verify completely whether the claims are actually validated and the correctness of the proofs.

**Weaknesses:**

* While I am not an expert in this field, I find the writing to be really opaque, right from the abstract. The opening words of the abstract are jargon and right until the last sentence I had no clear understanding of the problem being dealt with. This persists during the introduction as well, where I acknowledge that my comments might arise from my own ignorance and lack of expertise, but some sentences just sound like jargon to me when the whole problem could have been motivated in a much better way. Some other writing issues:
  * Line 81 undefined reference
  * References that don't exist – which is quite serious in my opinion – I was either not able to find the following papers or their links were undefined or both:
    1. Hayou et al. "On the impact of residual connections on the lipschitz constant of neural nets." In Advances in Neural Information Processing Systems, 2019 – does not exist
    2. Manakul et al. "Selfcheckgpt: Zero-resource black-box hallucination detection for generative large language models." – [provided link](https://aclanthology.org/2023.emnlp-main.722/) is incorrect
    3. N. Mündler and colleagues. "Self-contradictory hallucinations of large language models: Evaluation, detection and mitigation." In International Conference on Learning Representations (ICLR), 2024 – [provided link](https://openreview.net/forum?id=hgtX9Z8H6z) does not exist
    4. Kaiqing Yang, Guanghui Lan, and Tamer Basar. Learning deep mean field games for modeling large population behavior. In International Conference on Learning Representations (ICLR), 2018. URL https://openreview.net/forum?id=ryxY-pZAW – paper does not exist, but is a potentially hallucinated version of https://arxiv.org/abs/1711.03156.

  Given the above points, I seriously doubt the truthfulness of the LLM usage statement: "Large language models were used only for polishing language, fixing minor coding errors, and triaging related work. The proofs, analyses, and results are by the authors, and every cited reference was verified directly." Could the authors please clarify this?
* It seems that the reproducibility statement is also riddled with incorrect details. The [Colab link](https://colab.research.google.com/embedded/projects/prj-prd-data-learning-ddb6/locations/europe-west4/repositories/ce0ee1f7-19db-4562-b14f-52907a2e3e70) provided at the head of the file does not work and only the `neutrality_audit.py` script was provided. There is no repository or requirements files as claimed, nor is SciPy used as stated. It is unfortunate that a paper on hallucinations should be riddled with what are potentially the effects of LLM hallucinations as well. Again, I hope I have not misunderstood anything, but could the authors clarify this?
* The authors have only considered horizons of 32 in their experiments, which seems quite small.
* All models are only GPT-2 variants, there exist more modern LLMs with open weights/architectures that satisfy the architectural assumptions here. Furthermore, if the authors would want to actually compute the bounds derived in the paper, a much simpler setup could be considered (a toy model).
* The theoretical bounds derived seem to be loose and very conservative, based on my understanding.
* Some of the CIs in Table 2 are really wide. Could the authors comment on this?

**Questions:**

None aside from the points raised in the Weaknesses section.

**Details Of Ethics Concerns:**

I acknowledge that I am not an expert in this domain, but it seems like the paper has several details as pointed out in my Weaknesses section that could be the result of improper/undeclared and unverified LLM usage. Erring on the side of caution, I think this paper needs ethics reviews. I am unsure if this counts as a research integrity issue or responsible research issue, but my concern is that there are several inaccurate details in the paper that could be the result of poor research standards.

---

> ### Author Response · Authors · 2025-11-20
> **Part I (Referencing and Ethics)**
>
> We thank the reviewer for the time and care invested in examining the submission. We appreciate the diligence, and we apologise for the clerical mistakes that caused confusion. We are grateful that the reviewer engaged with the work, even while noting limited expertise in this specific area. The feedback is valuable and will be addressed in detail below.
>
> ${\textbf{Referencing and Ethics}}$
> > Given the above points, I seriously doubt the truthfulness of the LLM usage statement: "Large language models were used only for polishing language, fixing minor coding errors, and triaging related work. The proofs, analyses, and results are by the authors, and every cited reference was verified directly." Could the authors please clarify this?
>
> We want to clarify what happened and how it relates to the ICLR Code of Ethics and the ICLR 2026 guidance on the use of LLMs.
>
> We used an LLM at the very beginning only to triage an initial large set of related work, which is permitted under the ICLR 2026 policy when disclosed. After that triage step we manually checked every paper and constructed the reference list ourselves. The included cited works exist and are used in the correct scientific context.
>
> The incorrect URLs and altered titles entered the manuscript only at the very end. We asked an LLM to convert our manually prepared list into a BibTeX file (.bib) with clickable links. Only a few BibTeX entries via google scholar verification contained URLs, so to make reviewing easier we attempted to populate missing URL fields automatically.
>
> During this formatting step the LLM automatically filled the URL and DOI fields and introduced mistakes that we did not notice in the final pass. We did not recheck the autogenerated .bib after this step. This was a clerical oversight. The cited works, authors and titles correspond to real publications and were not invented.
>
> The ICLR guidance on LLM usage states that authors remain responsible for the accuracy of material produced by LLM tools and that the policy addresses the creation of nonexistent content. Our understanding is that this situation involves incorrect metadata rather than fabricated scientific content, and therefore does not fall under the types of ethics violations described in that guidance. We nevertheless apologise and take full responsibility for this clerical error.
>
> In the revision we will correct this by restoring and verifying every entry manually. For completeness, the referenced works are listed below, including the context in which they were used in the paper:
>
> $[1]$ https://proceedings.mlr.press/v97/hayou19a/hayou19a.pdf
>    Generalize the multiplicative stability property that Hayou established as the standard for ResNets, cited in Appendix
>
> $[2]$ https://www.amanchadha.com/research/2401.01313.pdf
>    Survey of hallucination mitigation techniques in LLMs, cited in Introduction
>
> $[3]$ https://arxiv.org/abs/2305.15852
>    Prompting based existing diagnostic framework for detection of hallucination, cited in Introduction
>
> $[4]$ https://arxiv.org/abs/1711.03156
> One of the first works that applies the mean field game formalism combined with deep neural networks for modelling large population behaviour, aligned with the structural scaling mean field lift perspective we adopt, cited in Relation to prior work

---

> ### Author Response · Authors · 2025-11-20
> **Part II (Reproducibility)**
>
> ${\textbf{Reproducibility}}$
>
> > It seems that the reproducibility statement is also riddled with incorrect details. The Colab link provided at the head of the file does not work and only the neutrality_audit.py script was provided. There is no repository or requirements files as claimed, nor is SciPy used as stated. It is unfortunate that a paper on hallucinations should be riddled with what are potentially the effects of LLM hallucinations as well. Again, I hope I have not misunderstood anything, but could the authors clarify this?
>
> We thank the reviewer for raising these concerns. The inconsistencies came from version control issues during final formatting. They do not affect the analysis, execution, or validity of the experiments.
>
> The submitted neutrality_audit.py is complete and self contained. Running this file reproduces the experiments and tables. Seeds, prompts, temperatures, sampling rules, and model names are stated explicitly. The script allows adjustments of parameters, but running the larger configurations is compute intensive. The file runs as a standard Python script because the header is a comment block. It installs dependencies, loads the models, runs the closed and open probe tests, the layer as agent tests, the pilot plan, the randomization checks, and writes results.csv and msgs.csv. The pilot plan is included only to manage compute and is not required by the theory.
>
> SciPy and JSON appeared only in earlier prototypes. The submitted script uses NumPy for the statistical power calculation (previous version was SciPy) and does not write JSON logs. We will correct these outdated references in the reproducibility text.
>
> The reproducibility statement also pointed to a Colab repository, hence the broken link in the file. This reflected our original plan to host the audit in Colab for free and easily accessible GPU execution. Unfortunately, a public Colab repository could not be anonymised. We therefore exported the full audit into a single Python file and submitted that file. The link will be removed.
>
> In the revision we will update the reproducibility statement and include a clear empirical testing protocol as requested by another reviewer.

---

> ### Author Response · Authors · 2025-11-20
> **Part III (Horizon)**
>
> ${\textbf{Horizon}}$
> > The authors have only considered horizons of 32 in their experiments, which seems quite small.
>
> We agree that the horizons in the experiments are modest. This choice was deliberate and not due to a limitation of the theory. The neutrality theorem in Section 3 is uniform in time. It states that in the regime where the operator norm assumptions hold,
> $$
> \mathbb{E}[D_{t+1} - D_t \mid \text{history}] = 0 \quad \text{for every } t.
> $$
>
> Short horizons were used for practical reasons. First, GPT2 medium and GPT2 large make long rollouts expensive. One neutrality step requires three arms, each expanded into $$\(M_{\text{SIB}} = 16\)$$ sibling continuations, so a single decoded token triggers \(48\) forward evaluations of GPT2 large. The computational cost grows linearly in the horizon. The theory itself has no horizon limit and longer rollouts are possible when compute permits.
>
> Second, the tests work on the sequence of drift increments. Statistical power comes from the number of pooled increments across prompts and seeds, not from very long single trajectories. Third, the closed and open probes are designed to detect systematic contraction or expansion through the sign of the mean increment. Any consistent bias would already be visible at these horizons.
>
> In the revision we will add a supplemental experiment on a smaller model such as distilgpt2 or GPT2 small using longer horizons, showing that the empirical behaviour remains neutral over extended rollouts, in line with the time uniform theorem.

---

> ### Author Response · Authors · 2025-11-20
> **Part IV (Theoretical Bounds and Model Agnosticism)**
>
> $\underline{\textbf{Theoretical Bounds and Model Agnosticism}}$
>
> > All models are only GPT-2 variants, there exist more modern LLMs with open weights/architectures that satisfy the architectural assumptions here. Furthermore, if the authors would want to actually compute the bounds derived in the paper, a much simpler setup could be considered (a toy model).
> > The theoretical bounds derived seem to be loose and very conservative, based on my understanding.
>
> We consider both points together. The bounds are conservative by design, and this is also the main difference from the prior literature. The existing works analyse isolated components such as attention blocks or residual connections under typical-input or average-case assumptions. These give intuition, but they are not roll-out level guarantees for paired autoregressive trajectories.
>
> Our bounds serve a different purpose. The neutrality theorem concerns the full autoregressive backbone, so the inequalities must hold for $\mathit{every}$ prompt, seed, horizon, and model size that satisfies the operator-norm assumptions. Because the result is global rather than layer-local, the bounds must cover the worst-case behaviour of the residual update and the decoded map: if they were tight only on typical inputs, the neutrality result would not follow.
>
> This is also why earlier bounds cannot be reused. Prior results provide layer-level Lipschitz estimates or heuristic stability arguments, but not the block-local operator-norm bounds or the drift decomposition needed in Section 3. Our proof requires an explicit predictable component of the drift increment so that the martingale term can be isolated and
> $$
> \mathbb{E}[D(t+1) - D(t) \mid \text{history}]
> $$
> can be shown to vanish. Achieving this requires uniform inequalities, which forces a conservative envelope.
>
> The numerical value of the upper bound itself could indeed be computed on a much simpler architecture. Any architecture that satisfies those assumptions inherits the same predictable-drift bound. Verifying it on a small toy network would therefore only reproduce this already established fact and would not probe the depth or scale-dependent behaviour. The more informative test is whether the theoretical mechanism predicted by the theorem appears in more realistic models, which is what Section 4 examines.
>
> In the revision we will add an additional test on a different LLM, within our compute budget, which was also requested by another reviewer.

---

> ### Author Response · Authors · 2025-11-20
> **Part V (Confidence Intervals)**
>
> $\underline{\textbf{Confidence Intervals}}$
>
> > Some of the CIs in Table 2 are really wide. Could the authors comment on this?
>
> The wide CIs in Table 2 reflect the fact that the layer-as-agent diagnostic is computed at the block level and is intentionally coarse. For each residual block, the diagnostic computes a finite-difference action value, and the reported mean, standard error, and 95% confidence interval are obtained by bootstrapping these block-level actions. Because only a small subset of blocks is examined, the bootstrap sample is correspondingly limited. This is what we note on page 7: “the intervals are wide, which reflects the limited sample size at this granularity, but the absence of systematic deviation suggests that no individual block introduces consistent bias”.
>
> The purpose of the layer as agent diagnostic is to provide a coarse consistency check at the block level. The audit can be run either at the output of the whole model or anchored earlier in the network. By anchoring at a specific residual block and repeating the same finite difference drift calculation at that point, we can see whether the behaviour observed at the final output is already present inside the model. This shows whether the neutrality mechanism is a global property of the autoregressive backbone or whether it depends on late-stage interactions only. The diagnostic is therefore not a separate statistical test, but a simple structural check that the block-anchored drift estimates still centre around zero when examined inside the network. The block-level results in Table 2 offer only a lower resolution internal view.

---

> > ### Comment · Reviewer_tE7k · 2025-11-22
> >
> > I thank the authors for their response. I do not see a new version of the paper yet, but here are my remarks after reading the responses (here and to other reviewers):
> > * I appreciate the authors' explanation about their use of LLMs, but it is quite concerning to me that the authors were willing to use the LLM output as-is without properly checking the result, while claiming that "every cited reference was verified directly" in their LLM use statement. In one case I mentioned, the entire author list was hallucinated. Overall, I appreciate the acknowledgement of the oversight from the authors, but hope this will serve as a serious, cautionary tale.
> > * My take as of now is that the paper would need significant rewriting to improve its clarity/communication, and it would thus need a whole new set of reviews that may be beyond the scope of the discussion period. If the authors are able to provide a new revision soon, I could take a look and offer comments, but I'm not sure there is enough time to properly re-evaluate the paper in its entirety.

---

> ### Author Response · Authors · 2025-11-25
> **Revision**
>
> Thank you very much for your willingness to revisit our manuscript. Regarding the earlier concern: the issue arose during an automated processing step that inserted links into our .bib file, and it was our responsibility to detect and correct those substitutions before submission. We again apologize for this oversight. We have now removed all incorrect links and manually verified the remaining references. Going forward, we will avoid automated changes to the .bib and ensure that every entry is checked directly.
>
> Since receiving the reviews, we have carried out a major clarity-oriented revision of the manuscript to address the concerns raised in the reviewer comments:
>
> • The abstract and sections 1-3 have been rewritten with clearer motivation.
>
> • Adjustments were made in the conclusion to improve clarity. Sections 4–6 and the appendices containing the proofs remain unchanged, except for a brief clarification in the empirical protocol introduction (beginning of Section 4).
>
> • A table of symbols and notation has been added to the appendix.
>
> • A clarification with respect to semantic/factual hallucinations has been included.
>
> • The revision now includes model-agnostic audits on Qwen 2.5 (0.5 to 3 billion parameters) using a lighter script. The block-based layer-as-agent analysis is not included for Qwen due to computational cost, and this limitation is clearly stated.
>
> Because the contribution of the paper is theoretical, the mathematical development necessarily remains detailed, but the exposition has been reorganized and clarified so that the structure of the argument and the role of each component are now presented more transparently.
>
> We are again sincerely grateful that you are willing to take another look at our manuscript within the limits of the discussion period.

---

### Official Review · Reviewer_rgqj · 2025-11-07

**Soundness:** 1
**Presentation:** 1
**Contribution:** 1
**Rating:** 0
**Confidence:** 4

**Summary:**

This paper's central claim appears to be that hallucinations in transformers are a 'persistent' and inherent property, suggesting interventions cannot eliminate them. The primary issue with this submission is its severe lack of clarity, which renders the paper inaccessible. The writing is extremely opaque.

To illustrate, the abstract alone introduces a deluge of specialized, undefined terms that make the paper's premise inaccessible. Just from the first few lines, a reader is forced to ask: What are "onset" and "persistence"? The paper says it "separates" them, but the "of what" (presumably hallucinations) is only implied. What does the "absence of contractive or expansive bias at the decoded level" mean in a practical, understandable sense? What are "corridor constants"? How can a "constant" be "falsifiable"? What are "open probes"? What is the "drift" being referred to? What is the "Neutrality" being proven?

These questions are merely a sample from the abstract; this systemic lack of clarity continues throughout the paper.

Due to these fundamental presentation flaws, I cannot provide a competent or fair evaluation of the paper's technical soundness.

In its current state, the work is not ready for publication at ICLR. It would require a complete and substantial rewrite to become understandable and reviewable. Therefore, I must recommend **strong rejection**.

**Strengths:**

N/A.

**Weaknesses:**

See above.

**Questions:**

N/A.

---

> ### Author Response · Authors · 2025-11-20
> **Response on soundness, contribution, and presentation**
>
> We thank the reviewer for the report. We agree that the presentation requires improvement. The revised version will introduce all concepts in a staged manner and remove terminology that is not necessary.
>
> The review states that the technical content could not be evaluated due to clarity issues. Yet the review assigns the lowest soundness and contribution scores with a confidence level of 4. These two axes are intended to measure technical correctness and originality. Using presentation issues to determine these scores is inconsistent with the rubric definitions used at (previous) major conferences. We rely on the standard interpretation of these axes, and we are not aware of any change in how these categories are intended to be understood.
>
> $\underline{\text{Soundness}}$: The submission provides complete definitions and proofs. The sequence from definitions to propositions and theorems can be checked directly. The code is provided as a .py file. While we again acknowledge that the presentation is opaque, the mathematical steps and proofs can be followed directly and allow a complete assessment of correctness.
>
> $\underline{\text{Contribution}}$: The work gives a mathematical description of how small internal deviations can persist during generation. These deviations are a structural requirement for any lasting semantic error, although the analysis itself is purely internal and does not evaluate semantic content.
> Prior work focuses on detecting or mitigating semantic errors. This paper instead studies how small internal changes evolve from one step to the next and shows that the model’s internal computations do not cause these changes to systematically shrink or grow in expectation. As a result, such deviations can persist across the generation process. This also provides a basis for analysing the stability of generated sequences even when intermediate states are not accessible. To our knowledge, this type of mathematical formulation has not been presented before.
>
> We will revise the presentation as outlined above and kindly ask the reviewer to revisit their evaluation once the revised version is provided.

---

### Author Response · Authors · 2025-11-20
**General remark and outline of our rebuttal process**

We thank all reviewers for the time and attention given to the evaluation. We will not submit all replies simultaneously. We will begin addressing the points now and will provide complete responses by the end of tomorrow.

Several reviews raised concerns about opacity and limited clarity. Rather than repeating this in each reply, we confirm that we will revise the entire manuscript to improve exposition, reduce jargon, and ensure clear presentation throughout.

After providing replies to all points, in which we will state the modifications to be made, we will then carry out these changes in the manuscript.

---

### Meta-Review · Area_Chair_azy1 · 2026-01-08

**Summary:**

All reviewers raised serious concerns about presentation and clarity, including the reviewer who gave a positive score (uH9y). The abstract and early sections are widely described as opaque and jargon-heavy, making it difficult to evaluate the main claims and contributions of this paper. Because of this, reviewers report limited ability to confidently assess technical soundness and significance within the review bandwidth available.

One reviewer (ZZth) additionally questions the paper's framing of hallucinations, arguing that the analysis is primarily about predictive/trajectory divergence and does not directly establish claims about semantic hallucination persistence.

Overall, the dominant issue affecting the evaluation is that the paper remains difficult to parse and evaluate.

**Reviewer Concerns:**

The authors made an effort in the rebuttal and revision to address clarity, such as rewriting the abstract and early sections.

However, the main concern remains. Even after the revision, the manuscript is still difficult to follow. Regarding the framing around hallucinations, while the authors provided clarifications of their claims, the analysis remains focused on predictive divergence rather than directly on hallucinations themselves.

**Reviewer Scores:**

As the main concerns have not yet been fully addressed, it is unlikely that the reviewers' scores would change.

---

### Decision · Program_Chairs · 2026-01-26

Reject